# WHEN GRAPH NEURAL NETWORKS MEET DYNAMIC MODE DECOMPOSITION

**Dai Shi** *

University of Sydney

**Lequan Lin**[*][†]

University of Sydney

**Andi Han**

Riken AIP

**Zhiyong Wang**

University of Sydney

**Yi Guo**

Western Sydney University

**Junbin Gao**

University of Sydney

## ABSTRACT

Graph Neural Networks (GNNs) have emerged as fundamental tools for a wide range of prediction tasks on graph-structured data. Recent studies have drawn analogies between GNN feature propagation and diffusion processes, which can be interpreted as dynamical systems. In this paper, we delve deeper into this perspective by connecting the dynamics in GNNs to modern Koopman theory and its numerical method, Dynamic Mode Decomposition (DMD). We illustrate how DMD can estimate a low-rank, finite-dimensional linear operator based on multiple states of the system, effectively approximating potential nonlinear interactions between nodes in the graph. This approach allows us to capture complex dynamics within the graph accurately and efficiently. We theoretically establish a connection between the DMD-estimated operator and the original dynamic operator between system states. Building upon this foundation, we introduce a family of DMD-GNN models that effectively leverage the low-rank eigenfunctions provided by the DMD algorithm. We further discuss the potential of enhancing our approach by incorporating domain-specific constraints such as symmetry into the DMD computation, allowing the corresponding GNN models to respect known physical properties of the underlying system. Our work paves the path for applying advanced dynamical system analysis tools via GNNs. We validate our approach through extensive experiments on various learning tasks, including directed graphs, large-scale graphs, long-range interactions, and spatial-temporal graphs. We also empirically verify that our proposed models can serve as powerful encoders for link prediction tasks. The results demonstrate that our DMD-enhanced GNNs achieve state-of-the-art performance, highlighting the effectiveness of integrating DMD into GNN frameworks.

## 1 INTRODUCTION

Graph Neural Networks (GNNs) (Kipf & Welling, 2017) have become fundamental tools for processing graph-structured data across a wide range of learning tasks. Recently, numerous studies (Thorpe et al., 2022; Chamberlain et al., 2021c; Wang et al., 2023; Choi et al., 2023) have established a strong connection between GNN feature propagation and diffusion processes. Building upon this connection, many of these works have integrated complex physical processes into graph diffusion to mitigate inherent computational issues in GNNs, such as over-smoothing (Rusch et al., 2023) and over-squashing (Shi et al., 2023b). While these approaches have yielded significant theoretical and empirical results from the feature propagation perspective, there remains a gap in analyzing these dynamics through the lens of physics, particularly from a dynamical systems viewpoint.

From the dynamical systems perspective, a carefully analyzed and refined dynamic can potentially enhance GNN performance by providing deeper insights into feature propagation over graphs.

---

*Equal contribution. Dai Shi is the corresponding author. ✉ dai.shi@sydney.edu.au.

†In memory of MeiMei 🐕 , whose love and companionship will always be remembered.

However, to tackle more challenging tasks, recent GNNs often adopt advanced physical dynamics (Han et al., 2024), increasing complexity and hindering interpretability. Thus, a well-established tool is needed to analyze these dynamics and provide deeper insights. Motivated by this, we delve deeper into the GNN dynamics by employing Dynamic Mode Decomposition (DMD) (Tu, 2013), a fundamental tool from fluid dynamics theory for approximating the Koopman operator (Koopman, 1931), which is essentially designed to approximate the non-linear dynamic using an (infinite-dimensional) linear operator. Specifically, DMD approximates the Koopman operator with finite-dimensional representations only via the states (i.e., *snapshots*) of the system, regardless of its complexity. In addition, DMD produces the so-called *DMD-modes*, which are a collection of refined eigenbases of the system's original operator (e.g., graph adjacency matrix) with potentially lower-dimensional. These modes capture the principal components driving the underlying GNN dynamics and offer a data-driven adjusted domain for spectral filtering. Building on these advantageous properties of DMD, in this work, we propose DMD-enhanced GNNs (DMD-GNNs). We show that our proposed DMD-GNNs not only can capture the characteristics of the original dynamics in a linear manner but also have the potential to adopt more complex tasks (e.g., long-range graphs (Dwivedi et al., 2022)) in which the original model of DMD-GNNs usually delivers weak performance.

**Contribution**   In this paper, we establish a novel link between feature dynamics in graph neural networks and the Koopman operator learning theory through DMD. We begin by formulating the dynamic system, Koopman operator, and DMD in Section 3 and show the link between GNN and dynamic system in Section 4. In Section 5, we introduce our newly designed data-driven graph DMD approach, demonstrating how it captures the principal components that drive the underlying complex physics on the graph. Accordingly, a family of DMD-GNNs can be smoothly derived from DMD with a flexible choice of initial dynamics. In Section 6, we analyze the properties of graph DMD from a dynamical systems perspective, showing that the manifold containing DMD outputs is locally topologically conjugate to a specific spectral submanifold tangent to the system origin. This reveals the underlying geometry between our graph DMD and the actual characteristics of the data. We further investigate how different choices of initial dynamics affect the range of the DMD output spectrum, which is essential for DMD-GNNs in fitting both homophily and heterophily graphs. In Section 7, we evaluate the performance of DMD-GNNs on various learning tasks, including node classification on citation networks, long-range datasets, and node regression on spatial-temporal graphs (STGs), along with tests of parameter sensitivities. We also show that DMD-GNNs can be powerful encoders for the link prediction tasks. Finally, in Section 8, we explore the potential of incorporating additional constraints on DMD and deploying physics-informed DMD for directed graphs, showing the model's potential of imitating complex physical dynamics, which leads to a range of future research directions at the intersection of DMD and GNNs.

## 2   RELATED WORKS

**Graph Neural Diffusion Models**   Many recent works have explored the link between the feature propagation in GNNs to the so-called physical diffusion process, leading to a variety of diffusion-based or physics-informed models (Chamberlain et al., 2021a; Choi et al., 2023; Shi et al., 2024a). By integrating complex physical dynamic systems into the graph domain, these methods often achieve superior performance across different learning tasks. However, the analysis of whether the benefits from incorporated dynamics are sufficiently leveraged in GNN is rarely conducted and can only be observed in the recent works (Wang et al., 2023). Furthermore, although they are data-driven, attention-based GNNs (e.g., GAT(Veličković et al., 2018)) and transformer-based diffusion models (e.g., Difformer(Wu et al., 2023a)) compute node-level attention scores based on feature similarities within each layer (i.e., the current system state) but rarely consider cross-layer feature interactions, which could better capture relationships between multiple system states. In this work, we analyze some commonly implemented GNNs and their dynamics through DMD, showing that a low-rank estimate of the system original operator from multiple system states is sufficient to induce identical or even better learning performances. Our theoretical results establish a bridge between DMD and the operators of the underlying dynamics.

**Koopman Operator and Dynamic Mode Decomposition**   The Koopman operator theory (Koopman, 1931) has become one of the most widely applied tools for analyzing modern dynamical systems (Brunton et al., 2021). Fundamentally, Koopman theory posits that any finite-dimensional nonlinear

dynamic can be studied as an infinite-dimensional linear operator, denoted as $\mathcal{K}$. To identify finite-dimensional representations of $\mathcal{K}$, numerous numerical methods have been developed, with DMD emerging as one of the most popular. DMD has become a leading tool in aerodynamics and fluid physics (Azencot et al., 2020; Eivazi et al., 2021). From a machine learning perspective, Koopman autoencoders (Nayak et al., 2024) have been developed to learn both the system's measurement functions and the operator simultaneously. Furthermore, the spectral characteristics (i.e., eigenvalues and eigenvectors) of the estimated Koopman operator have been leveraged in recent advancements in time series analysis (Berman et al., 2023; Liu et al., 2024; Lange et al., 2021). Theoretically, the work by Baddoo et al. (2023) extends the DMD algorithm to incorporate more information about the underlying physics (e.g., symmetry, low-rank structures) into its outputs, resulting in a suite of physics-informed DMD methods. Additionally, (Haller & Kaszás, 2024) proved that the estimates computed from DMD belong to the slow decay submanifold that is tangent to the slow-spectral subspaces, leading to a deeper understanding of the DMD outputs and the underlying physics.

# 3 DYNAMIC SYSTEMS, KOOPMAN OPERATOR AND DMD

We consider an autonomous dynamic system which can be defined as:

$$\dot{\mathbf{x}} = \mathbf{f}(\mathbf{x}), \qquad \mathbf{x} \in \mathbb{R}^d, \qquad \mathbf{f} \in \mathcal{C}^1(\mathbb{R}^d), \tag{1}$$

in which $\mathbf{x} = \mathbf{x}(t)$ is a $d$-dimensional vector serving as the state of the system at $t$. An evolution or trajectory of the system $\{\mathbf{x}(t)\}_{t \in \mathbb{R}_+}$ can induce a so-called flow mapping $\mathbf{F}_t : \mathbb{R}^d \to \mathbb{R}^d$ that takes the state $\mathbf{x}(0)$ at time 0 to the state $\mathbf{x}(t)$ at time $t > 0$.

In 1931, Bernard O. Koopman demonstrated that it is possible to represent a nonlinear dynamical system in equation (1) by using an infinite-dimensional linear operator that acts on a Hilbert space of measurement functions of the state $\mathbf{x}$ (Koopman, 1931). Specifically let us consider the smooth functions: $\phi(\mathbf{x}) \in \mathcal{C}^1(\mathbb{R}^d, \mathbb{R}^m)$, where $\phi$ and $\phi(\mathbf{x})$ are often named as the measurement function and observations, respectively, (or interchangeably named as observables in some works (Schmid, 2022)). The Koopman operator $\mathcal{K}_t$ is an infinite-dimensional linear operator that maps a measurement function $\phi$ into a new function $\mathcal{K}_t\phi$ such that its function values at a state $\mathbf{x}$ (imagining it is at time 0) is given by $\phi$'s function value at the state given by the flow map $\mathbf{F}_t$ after time $t$, i.e.,

$$(\mathcal{K}_t\phi)(\mathbf{x}) = \phi(\mathbf{F}_t(\mathbf{x})), \quad \forall \mathbf{x} \in \mathbb{R}^d. \tag{2}$$

This can be written as $\mathcal{K}_t\phi = \phi \circ \mathbf{F}_t$. It is easy to prove that, for any two measurement functions $\phi_1$ and $\mathcal{K}_t(a\phi_1 + b\phi_2) = a\mathcal{K}_t\phi_1 + b\mathcal{K}_t\phi_2$ where $a, b \in \mathbb{R}$. When considering all the possible advancing time $t$ along the flow maps, one can prove that $\{\mathcal{K}_t\}_{t \in \mathbb{R}_+}$ forms a semi-group which has an infinitesimal generator $\mathcal{K}$. In literature, this generator is called the Koopman operator of the underlying dynamical system, which satisfies the Koopman dynamical system: $\frac{d}{dt}\phi = \mathcal{K}\phi$.

For our purpose, we are more interested in the discrete nonlinear system defined by

$$\mathbf{x}_{k+1} = \mathbf{F}(\mathbf{x}_k). \tag{3}$$

The discrete system equation (3) itself defines a discrete path $\mathcal{X} = \{\mathbf{x}_k\}$. In the same merit of equation (2), the Koopman operator $\mathcal{K}$ for the nonlinear discrete system can be defined as

$$\mathcal{K}\phi(\mathbf{x}_k) = \phi(\mathbf{F}(\mathbf{x}_k)) = \phi(\mathbf{x}_{k+1}). \tag{4}$$

That is the value of the mapped function $\mathcal{K}\phi$ at the state in step $k$ is given by the value of measurement function $\phi$ at the state in the next along the path $\mathcal{X}$. As $\mathcal{K}$ is an infinite-dimensional linear operator in the measurement function space, this paves the way such that we analyze the underlying discrete nonlinear dynamical system in measurement space.

With a given set of finite number of observed data $\mathbf{\Phi}(\mathbf{X}(k)) = \{\phi(\mathbf{x}_k)\}$, the Dynamic Mode Decomposition (DMD), as one of the most popular numerical methods for approximating the Koopman operator, originally seeks for a finite-dimensional linear operator $\mathbf{K}$, such that

$$\mathbf{\Phi}(\mathbf{X}(k+1)) = \mathbf{K}\mathbf{\Phi}(\mathbf{X}(k)) \tag{5}$$

where $\mathbf{\Phi}(\mathbf{X}(k+1))$ is the dataset of one step shift version of dataset $\mathbf{\Phi}(\mathbf{X}(k))$. DMD and its variants have been widely deployed via various fields such as fluid dynamics (Arbabi & Mezic, 2017; Mezić, 2013), time series forecasting (Liu et al., 2024), environmental engineering (Li et al., 2023), and solving partial differential equations (Kutz et al., 2016b)

## 4  HOW GNNS RESONATE WITH DYNAMIC SYSTEMS

Throughout this paper, we denote $\mathcal{G}(\mathcal{V}, \mathcal{E})$ as graphs that can be weighted and directed, where $\mathcal{V}$ and $\mathcal{E}$ are the set of nodes and edges. Consider a multilayer GNN on the graph $\mathcal{G}(\mathcal{V}, \mathcal{E})$, and let $\mathbf{H}(\ell) \in \mathbb{R}^{N \times d_\ell}$ be the node feature matrix at layer $\ell$, and $\mathbf{A}$ and $\mathbf{L} \in \mathbb{R}^{N \times N}$ be the adjacency and Laplacian matrices of the graph, respectively. In general, most GNNs propagate the node signals through either spatial aggregation or spectral filtering (Wu et al., 2020), denoted as:

$$\mathbf{H}(\ell + 1) = \mathcal{F}(\mathbf{H}(\ell)), \tag{6}$$

where one can have $\mathbf{H}(0) = \mathbf{X}$, the original feature matrix or $\mathbf{H}(0) = \mathbf{X}\mathbf{W}$, and the generic function $\mathcal{F}$ maps node features from one layer to another. When $\mathcal{F}(\mathbf{H}) = \mathbf{A}\mathbf{H}$ or its variants, e.g., $\mathbf{A}$ is reweighed based on the attention mechanism or rewired according to the graph topology, equation (6) represents those spatial GNNs (Kipf & Welling, 2017; Veličković et al., 2018). On the other hand, if $\mathcal{F}(\mathbf{H}) = \mathbf{U}\mathrm{diag}(\boldsymbol{\theta})\mathbf{U}^\top \mathbf{H}$ or its variants, e.g., $\mathcal{F}(\mathbf{H}) = \mathbf{U}\sin^2(\mathrm{diag}(\boldsymbol{\theta}_1))\mathbf{U}^\top \mathbf{H} + \mathbf{U}\cos^2(\mathrm{diag}(\boldsymbol{\theta}_2))\mathbf{U}^\top \mathbf{H}$, then equation (6) becomes those spectral or multi-scale GNNs (Han et al., 2022). In addition, it is well-studied that many GNN dynamics in equation (6) are equivalent to the discretized version of some continuous diffusion processes (Han et al., 2024). Specifically, let $\mathbf{h}_i \in \mathbb{R}^d$ be the signal over node $i$, the graph diffusion process can be denoted as:

$$\frac{\partial \mathbf{h}_i(t)}{\partial t} = \sum_{j:(i,j)\in\mathcal{E}} \mathcal{S}_{i,j}(\mathbf{h}_i(t), \mathbf{h}_j(t), t)(\mathbf{h}_j(t) - \mathbf{h}_i(t)), \tag{7}$$

where the function $\mathcal{S}_{i,j}(\mathbf{h}_i(t), \mathbf{h}_j(t), t) : \mathbb{R}^d \times \mathbb{R}^d \times [0, \infty) \to \mathbb{R}$ defines the diffusivity coefficient controlling the diffusion strength pariwisely for any given $t$. In the simplest case, when $\mathcal{S}_{i,j}(\mathbf{h}_i, \mathbf{h}_j, t) = 1, \forall (i, j) \in \mathcal{E}$ the diffusion process can be written as $\frac{\partial \mathbf{h}_i}{\partial t} = \mathrm{div}(\nabla \mathbf{H})_i = -(\mathbf{L}\mathbf{H})_i$, in which we denote $\mathrm{div}$ and $\nabla$ as the divergence and gradient operator on the graph. The solution of the diffusion equation is given by the so-called heat kernel, i.e., $\mathbf{X}(t) = \exp(-t\mathbf{L})\mathbf{H}(0)$, suggesting an exponential decay of the feature dissimilarities known as over-smoothing (OSM) (Rusch et al., 2023). Accordingly, GNNs are required to find the balance between the diffusion and reaction processes, with the former homogenizes (smoothing) and later differentiates (sharpening) node features. One way to achieve this goal is to assign a certain source term to the diffusion process in equation (7), yielding the following:

$$\frac{\partial \mathbf{h}_i(t)}{\partial t} = \alpha \sum_{j:(i,j)\in\mathcal{E}} \mathcal{S}_{i,j}(\mathbf{h}_i(t), \mathbf{h}_j(t), t)(\mathbf{h}_j(t) - \mathbf{h}_i(t)) + \beta\, \mathcal{R}(\mathbf{h}_i(t), t), \tag{8}$$

where we let $\mathcal{R} : \mathbb{R}^d \times [0, +\infty) \to \mathbb{R}^d$ be a reaction function that acts only on the node features and $\alpha, \beta \in \mathbb{R}_+$ are hyperparameters to balance two terms. One can verify that when the system is dominated by the diffusion term, e.g., large $\alpha$ and small $\beta$, the corresponding GNNs will have more smoothing effects on the node features, whereas when the reaction term is dominated, node features tend to be dissimilar to each other, see (Han et al., 2022; Giovanni et al., 2023; Choi et al., 2023) for more detailed analysis. Furthermore, let $\alpha = \beta = 1$, when $\mathcal{R}(\mathbf{h}_i(t), t) = \mathbf{h}_i(t) \odot (1 - \mathbf{h}_i(t))$, we reach the Fisher information of the node features (Fisher, 1937), and if $\mathcal{R}(\mathbf{h}_i(t), t) = \mathbf{h}_i(t) \odot (1 - \mathbf{h}_i(t) \odot \mathbf{h}_i(t))$, then we have the source term presented as Allen-Cahn term (Wang et al., 2023). We refer to equation (10) in (Choi et al., 2023) and Appendix B for more examples. In practice, equation (8) can be implemented by the so-called $\mathrm{MLP_{in}}$ and $\mathrm{MLP_{out}}$ paradigm as:

$$\mathbf{H}(0) = \mathrm{MLP_{in}}(\mathbf{X}), \quad \mathbf{H}(T) = \mathbf{H}(0) + \int_0^T \mathcal{P}(\mathbf{H}(t))dt, \quad \mathbf{Y} = \mathrm{MLP_{out}}(\mathbf{H}(T)), \tag{9}$$

in which we let $\mathcal{P}(\mathbf{H}(t)) = -\alpha \mathbf{L}\mathbf{H}(t) + \beta \mathcal{R}(\mathbf{H}(t), t)$. Fixing $\alpha = \beta = 1$, then its Euler discretization yields $\mathbf{H}(\ell+1) = \mathbf{H}(\ell) - \mathbf{L}\mathbf{H}(\ell) + \mathcal{R}(\mathbf{H}(\ell), \ell) = \mathbf{A}\mathbf{H}(\ell) + \mathcal{R}(\mathbf{H}(\ell), \ell)$, yielding a generalized version of equation (6). Finally, if $\mathcal{R}(\mathbf{H}(\ell), \ell) = \mathbf{R}\mathbf{H}(\ell)$ where $\mathbf{R}$ is any matrix, then the diffusion and reaction terms can be combined as $(\mathbf{A} + \mathbf{R})\mathbf{H}(\ell)$, resulting a linear $\mathcal{F}$, i.e., $\mathcal{F}(a\mathbf{h}_i + b\mathbf{h}_j) = a\mathcal{F}(\mathbf{h}_i) + b\mathcal{F}(\mathbf{h}_j)$. Otherwise, in the case, e.g., $\mathcal{R}(\mathbf{H}(\ell), \ell) = \mathbf{H}(\ell) \odot (\mathbf{I} - \mathbf{H}(\ell) \odot \mathbf{H}(\ell))$, where the diffusion and reaction terms can not be combined, resulting as a non-linear feature propagation.

Clearly, equation (6) can be seen as a special case of discrete dynamical system equation (3). Similarly, the diffusion processes such as equation (7) and equation (8) are described as the continuous dynamical system equation (1). These analogies show the potential of linking the Koopman operator induced under the settings of equation (3) or equation (1) to the GNN dynamics. In the next section, we will show how the Koopman operator theory and its numerical approximation method (i.e., DMD) can facilitate analyzing GNN dynamics.

# 5 DYNAMIC MODE DECOMPOSITION IN GNNS

## 5.1 THE DATA-DRIVEN DMD ALGORITHM

Our primary goal is to leverage DMD to estimate a best-fitted finite-dimensional matrix $\mathbf{K}$ to approximate the infinite-dimensional Koopman operator $\mathcal{K}$. In the meantime, we also expect the process of estimating $\mathbf{K}$ can capture sufficient graph information by considering multiple rather than individual states of the system. Let's assume $\mathbf{H} = \mathbf{\Phi}(\mathbf{X})$, i.e., the graph features are the actual observation induced from $\mathbf{\Phi}$, and denote the operator $\mathcal{F}$ such that $\mathbf{H}(\ell+1) = \mathcal{F}\mathbf{H}(\ell)$, thus we have

$$\mathcal{F} = \mathbf{H}(\ell+1)\mathbf{H}(\ell)^{\dagger}, \tag{10}$$

where we let $\mathbf{H}(\ell)^{\dagger}$ be the pseudo-inverse of $\mathbf{H}(\ell)$. To show how DMD approximate $\mathcal{F}$, we start with the singular value decomposition (SVD) of $\mathbf{H}(\ell)$: $\mathbf{H}(\ell) = \mathbf{M\Sigma V}^*$, where we denote the upper case $*$ as the conjugate transpose or Hermitian, and the matrix $\mathbf{M}$, $\mathbf{\Sigma}$ and $\mathbf{V}^*$ are with the size of $N \times N$, $N \times d$ and $d \times d$, respectively. We note that given $N$ is larger than $d$ in most of cases (e.g., $\mathbf{X}$ in Cora is with the size $2708 \times 1433$), the rank of $\mathbf{H}(\ell)$ is at most $d$. Accordingly, we have

$$\mathcal{F} = \mathbf{H}(\ell+1)\mathbf{V\Sigma}^{-1}\mathbf{M}^*, \tag{11}$$

Furthermore, multiplying $\mathbf{M}^*$ and $\mathbf{M}$ on the left and right side of $\mathcal{F}$ yields

$$\mathbf{K} = \mathbf{M}^*\mathcal{F}\mathbf{M} = \mathbf{M}^*\mathbf{H}(\ell+1)\mathbf{V\Sigma}^{-1}, \tag{12}$$

suggesting a projection of $\mathcal{F}$ onto the column space of $\mathbf{M}^*$. Let the actual rank of $\mathbf{H}(\ell) = r$, then equation (12) directly suggests that in practice, to estimate the unknown $\mathcal{F}$ from system states, i.e., $\mathbf{H}(\ell)$ and $\mathbf{H}(\ell+1)$, one only need to focus on the $\mathbf{K}$ which is with reduced dimensionality of $r \times r$ which is obtained by $\min(N, d)$.

Denoting the eigendecomposition of $\mathbf{K}\mathbf{U}(\mathbf{K}) = \mathbf{U}(\mathbf{K})\mathbf{\Lambda}(\mathbf{K})$ since the rank of $\mathbf{H}(\ell)$ is $r$, therefore the rank of $\mathbf{K}$ is also $r$. Accordingly, one can check that the first $r$ eigenvalues of $\mathbf{K}$ are equal to the original $\mathcal{F}$, whereas the rest $N - r$ eigenvalue of $\mathbf{K}$ is zero. The corresponding (projected) *DMD modes* [1] can be denoted as the columns of $\mathbf{\Psi} = \mathbf{M}\mathbf{U}(\mathbf{K}) = \mathbf{H}(\ell)\mathbf{V\Sigma}^{-1}\mathbf{U}(\mathbf{K}) \in \mathbb{R}^{N \times r}$. One can verify that if the columns of $\mathbf{H}(\ell+1)$ are spanned by those of $\mathbf{H}(\ell)$, then the DMD modes and eigenvalues obtained from the eigendecomposition of $\mathbf{K}$ are the eigenvectors and eigenvalues of the operator $\mathcal{F}$ defined in equation (10) Accordingly, DMD provides a data-driven low-dimensional estimation of the operator $\mathcal{F}$, with both eigenvalues and eigenvectors based on $\mathbf{H}(\ell)$ and $\mathbf{H}(\ell+1)$.

## 5.2 DMD-GNNS AND FLEXIBLE CHOICE OF UNDERLYING PHYSICS

The illustration of DMD from the previous section suggests that to deploy DMD, one needs to assign an initial feature dynamic on the graph so that the snapshots (i.e., $\mathbf{H}(\ell)$ and $\mathbf{H}(\ell+1)$) can be collected. Through this paradigm, DMD can produce $\mathbf{K}$ as a low-rank estimation for approximating the original non-linear dynamics. That is

$$\mathbf{H}(\ell+1) = \mathcal{F}\mathbf{H}(\ell) \approx \mathbf{K}\mathbf{H}(\ell), \tag{13}$$

in which we note that the dimension of $\mathbf{K}$ is $N \times N$, however both $N - r$ eigenvectors and eigenvalues of $\mathbf{K}$ are all zeros based on equation (12). One can also leverage DMD via different feature dynamics such as oscillation (Rusch et al., 2022), wave equation (Eliasof et al., 2021) and quantum processing (Verdon et al., 2019) to unleash the underlying physics. More importantly, DMD provides the estimated dynamic modes and eigenvalues based on the data matrices from the initial dynamics, therefore, the feature propagation via the corresponding DMD-GNNs can expressed as a spectral filtering process by leveraging the estimated $\mathbf{\Psi}$, that is

$$\mathbf{H}(\ell+1) = \mathbf{\Psi}\mathrm{diag}(\boldsymbol{\theta})\mathbf{\Psi}^{\top}\mathbf{H}(\ell)\mathbf{W}(\ell), \tag{14}$$

where we denote $\mathbf{W}(\ell)$ as the learnable matrix for channel-mixing. In practice, such spectral filtering process can be further approximated via polynomial approximation, e.g., via the first $r$ eigenvalues

---

[1]One can also denote the so-called *exact DMD modes* by $\mathbf{\Psi} = \mathbf{H}(\ell+1)\mathbf{V\Sigma}^{-1}\mathbf{U}(\mathbf{K})$, and verify that if $\mathbf{H}(\ell)$ and $\mathbf{H}(\ell+1)$ shares the same column space, then the projected DMD modes will be equal to the exact DMD modes. Accordingly, in this work, we use the term *DMD modes* for simplicity reasons.

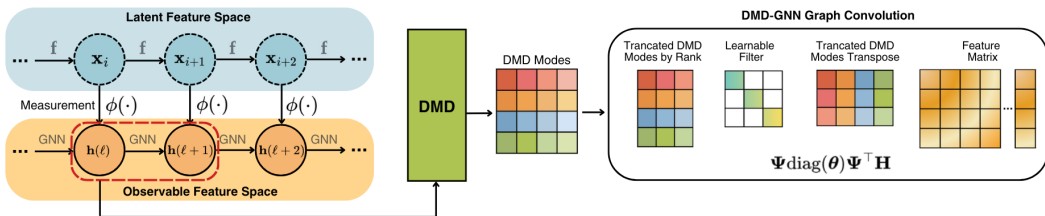

Figure 1: Illustration on how DMD-GNNs are designed.

(Defferrard et al., 2016). We further note that DMD also provides a flexible truncation rate (Ichinaga et al., 2024). Specifically, a rate $\xi \in [0, 1]$ is assigned to control the rate of truncation via the SVD process in DMD, e.g., $\xi = 0.85$ means DMD will select the singular values (from the largest) that are needed to reach 85% of the total spectral energy (i.e., $\sum_i \mathbf{\Lambda}_{ii}^2$). Accordingly, a higher value of $\xi$ indicates more eigenvalues are considered in $\mathbf{K}$, suggesting a wider range, including both higher and lower frequency components for spectral filtering in equation (14). On the other hand, in the case when $\xi$ is smaller, only those lower frequency components are preserved. Nevertheless, we found that practically setting $\xi \in [0.7, 0.9]$ is flexible enough for letting DMD-GNNs fit most of the graph benchmarks, see Section 6 and 7.4 for more theoretical and empirical justifications.

**Summary of the Model Procedure**   We visualize the steps on how our DMD-GNNs are designed in Figure 1. Initially, the underlying graph features $\mathbf{x}$ inside the top left blue circles are measured by the measurement function $\mathbf{\Phi}$, leading to the observed features $\mathbf{h}$ (in the left bottom orange cycles), which is further propagated by the initial GNNs to supply the inputs (*snapshots*) to DMD. DMD then takes these snapshots, e.g., $\mathbf{h}(\ell)$ and $\mathbf{h}(\ell+1)$, to produce the low-rank estimation of the GNN dynamics, namely DMD modes. After a customized truncation, the DMD modes (e.g., $\mathbf{\Psi} \in \mathbb{R}^{N \times r}$) are leveraged via DMD-GNNs to propagate node features via spectral filtering.

## 5.3   TOWARDS LEVERAGING MULTIPLE STATES VIA SPATIAL-TEMPORAL GRAPHS (STG)

Recall that the DMD algorithm produces a low-rank estimate of the system operator via multiple snapshots of the system, causing its wide application in time series forecasting (Kutz et al., 2016a). In terms of graph-based forecasting tasks, this can correspond to the node-level regression task via STGs. Specifically, the task can be interpreted as predicting the future time series window $\mathbf{H}_f = \{\mathbf{h}_{t_0+1}, \mathbf{h}_{t_0+2}, ..., \mathbf{h}_{t_0+f}\}$ given the past context window $\mathbf{H}_c = \{\mathbf{h}_{t_0-c+1}, \mathbf{h}_{t_0-c+2}, ..., \mathbf{h}_{t_0}\}$, where $f$ and $c$ are the length of future and past windows, by considering GNNs as a solver for the approximating the transaction between them (Bui et al., 2022; Lin et al., 2024), that is $\{\mathbf{h}_{t_0+1}, \mathbf{h}_{t_0+2}, ..., \mathbf{h}_{t_0+f}\} \approx \text{GNN}(\{\mathbf{h}_{t_0-c+1}, \mathbf{h}_{t_0-c+2}, ..., \mathbf{h}_{t_0}\})$, in which the graph structure, usually as prior knowledge in STGs, is leveraged consistently via the GNN dynamics. However, it is preferable to have the graph structure reweighed (Wu et al., 2023a) or rewired (Ali et al., 2022) especially when the prediction task is highly time-dependent, e.g., the traffic flow (speed) varies between rush hours and other time intervals. In this case, DMD shares a similar motivation by offering a data-driven estimated eigenspace for spectral filtering based on the selected initial dynamics and snapshots, resulting in a dynamic STG forecasting paradigm (Ivanovic & Pavone, 2019).

## 6   DMD RECOVERS THE UNDERLYING GEOMETRY

Recall that in Section 5.1 we briefly showed the relationship between the DMD modes and eigenvalues in $\mathbf{K}$ and those in the generic $\mathcal{F}$. In this section, we provide further analysis to show that

> *DMD captures the principal slow decay spectral subspace that is spanned by the $d$ basis vectors of the underlying dynamic system defined in equation (1).*

Our analysis adopts the settings in (Haller & Kaszás, 2024). First, without loss of generality, we assume that the underlying dynamic system contains a fixed point at the origin, i.e., $\mathbf{f}(\mathbf{0}) = \mathbf{0}$. We will also assume that the measurement function $\mathbf{\Phi} = \text{id}$ e.g., $\mathbf{\Phi}(\mathbf{X}) = \mathbf{X}$. Accordingly, one can

rewrite the dynamic in equation (1) as

$$\dot{\mathbf{x}} = \mathcal{A}\mathbf{x} + \widetilde{\mathbf{f}}(\mathbf{x}), \quad \mathcal{A} = D\mathbf{f}(\mathbf{0}), \quad \widetilde{\mathbf{f}}(\mathbf{x}) = o(|\mathbf{x}|), \quad \mathbf{x} \in \mathbb{R}^d, \tag{15}$$

where we denote $o(|\mathbf{x}|)$ to show that $\mathrm{Lim}_{\mathbf{x} \to \mathbf{0}} \left[ \frac{|\widetilde{\mathbf{f}}(\mathbf{x})|}{|\mathbf{x}|} \right] = 0$, indicating that $|\widetilde{\mathbf{f}}(\mathbf{x})|$ goes faster to $0$ than $|\mathbf{x}|$ near the origin. We also let $D\mathbf{f}(\mathbf{0})$ be the Jacobian matrix of the function $\mathbf{f}(\mathbf{x})$ when $\mathbf{x} = \mathbf{0}$. In addition, we assume that $\mathcal{A}$ is symmetric and the spectrum of $\mathcal{A}$ can be partitioned at a specific index (set as $d$ for simplicity). That is $\lambda_N \leq \cdots \lambda_{d+1} \leq \lambda_d \leq \cdots \lambda_1 < 0$, suggesting the existence of $d$-dimensional, normally attracting slow-spectral subspace $S = \mathrm{span}\{\mathbf{e}_1, \cdots, \mathbf{e}_d\}$, where we denote $\mathbf{e}$ be the basis vector. Similarly, one can denote the corresponding fast-decay subspace denoted as $F = \mathrm{span}\{\mathbf{e}_{d+1}, \cdots, \mathbf{e}_N\}$. We further let $\mathbf{U}(\mathcal{A}) = [\mathbf{U}(\mathcal{A})_S, \mathbf{U}(\mathcal{A})_F]$ be the corresponding eigenvectors in $\mathcal{A}$, with $\mathbf{U}(\mathcal{A})_S \in \mathbb{R}^{N \times d}$, $\mathbf{U}(\mathcal{A})_F \in \mathbb{R}^{N \times (N-d)}$. Then we have the following conclusion.

**Lemma 1** (Informal). *With mild conditions, further assume that* $\mathrm{rank}(\mathbf{H}) = d$ *and* $|\mathbf{U}(\mathcal{A})_F \mathbf{X}(\ell)|, |\mathbf{U}(\mathcal{A})_F \mathbf{X}(\ell+1)| \leq |\mathbf{U}(\mathcal{A})_S \mathbf{X}(\ell)|^{1+\tau}$ *for some* $\tau \in (0, 1]$*, suggesting in the underlying dynamics the subspace outside* $S$ *can quickly shrink out. Then DMD can capture the principal component that drives the underlying dynamic. Specifically, let the output of DMD be denoted as* $\mathcal{D}$ *(e.g.,* $\mathbf{K} \subseteq \mathcal{D}$*), then* $\mathcal{D}$ *is with the form*

$$\mathcal{D} = \mathbf{U}(\mathcal{A})_S e^{\mathbf{\Lambda}_S \Delta t} \mathbf{U}(\mathcal{A})_S^{\top} + \mathcal{O}(|\mathbf{U}(\mathcal{A})_S \mathbf{X}(\ell)|^{\tau}), \tag{16}$$

*and* $\mathcal{D}$ *is locally topologically conjugated with order* $\mathcal{O}(|\mathbf{U}(\mathcal{A})_S \mathbf{X}(\ell)|^{\tau})$ *error to the linearized dynamic on a* $d$*-dimensional, slow attracting spectral submanifold* $\mathcal{M}(S) \in \mathcal{C}^1$ *tangent to* $S$*.*

Proof and list of conditions for inducing Lemma 1 in Appendix A.1, equation (16) directly shows that DMD captures the principal component (i.e., $\mathbf{\Lambda}_s$) that drives the dynamic system from its spectral subspace. We further remark that the results of Lemma 1 can be smoothly applied to the discretized dynamic by assuming the spectrum range of $\mathcal{A}$ to be $|\lambda_N| \leq \cdots \leq |\lambda_{d+1}| \leq |\lambda_d| \leq \cdots |\lambda_1| < 1$, similarly, we show more detailed clarification in Appendix A.1.

**Guidance for GNN Dynamic, Parameter Initialization and Heterophily Adaptation** The conclusion in Lemma 1 provides guidance to select the initial dynamic and DMD-GNN filtering parameter initialization. For example, let's consider two different initial dynamics $\mathbf{H}(\ell+1) = \widehat{\mathbf{A}}\mathbf{H}(\ell)$ and $\mathbf{H}(\ell+1) = \widehat{\mathbf{A}}^2 \mathbf{H}(\ell)$, where we let $\widehat{\mathbf{A}} \in \mathbb{R}^{N \times N}$ be the normalized adjacency matrix. From the spectral graph theory, we have the eigenvalues of $\widehat{\mathbf{A}}$ (denoted as $\mathbf{\Lambda}(\widehat{\mathbf{A}})$) ranging from $-1$ to $1$. The solutions of the corresponding continuous dynamics of them are

$$\mathbf{H}(t) = e^{-(\mathbf{I} - \mathbf{\Lambda}(\widehat{\mathbf{A}}))t} \mathbf{H}(0) \quad \text{and} \quad \mathbf{H}(t) = e^{-(\mathbf{I} - \mathbf{\Lambda}^2(\widehat{\mathbf{A}}))t} \mathbf{H}(0), \tag{17}$$

and it is well-known that the eigenvalues of normalized Laplacian $\widehat{\mathbf{L}}$ is with the range of $[0, 2]$. However, one can verify that the range of the eigenvalues in $\mathbf{I} - \mathbf{\Lambda}^2(\widehat{\mathbf{A}})$ is $[0, 1]$ Therefore, with a fixed dimension of the slow decay subspace (i.e., fixed truncation rate), the second dynamic provides a smaller range of the eigenvalues and this is particularly useful for the graphs that require feature smoothing rather than sharpening dynamic in GNN, e.g., homophily graphs. Similarly, one can always bring the variation back by adding the source terms. For example, let the initial dynamic as $\mathbf{H}(\ell+1) = \widehat{\mathbf{A}}\mathbf{H}(\ell) + \mathbf{H}(\ell) = (\widehat{\mathbf{A}} + \mathbf{I})\mathbf{H}(\ell)$, and the solution of the continuous system is $\mathbf{H}(t) = e^{-\mathbf{\Lambda}(\widehat{\mathbf{A}})}\mathbf{H}(0)$, as $\mathbf{\Lambda}(\widehat{\mathbf{A}}) \in [-1, 1]$, hence corresponding slow decay subspace is back to the wider range, suggesting a potential mixed dynamic i.e., negative eigenvalues for sharpening and positive eigenvalues for smoothing, which benefits DMD-GNNs for fitting heterophily graphs.

## 7 EXPERIMENTS

We apply the proposed DMD GNNs to various learning tasks: 1) Node classification on both homophilic and heterophilic graphs; 2) Node classification on long-range graphs (Dwivedi et al., 2022); 3) spatial-temporal dynamic predictions; 4) As an efficient encoder for link prediction tasks. We conduct the parameter sensitivity analysis to test the model's sensitivity on the truncation rate $\xi$. We include detailed experiment setting information, and some further interpretation of our results via e.g., epidemiology and physics in Appendix C. All experiments are conducted in Python 3.11 on one NVIDIA®RTX 4090 GPU with 16384 CUDA cores and 24GB memory size.

Table 1: Node classification results in homophilic and heterophilic graphs, with the best highlighted in **bold**

| Methods | Cora | Citeseer | Pubmed | Texas | Wisconsin | Cornell | OGB-arXiv |
|---|---|---|---|---|---|---|---|
| MLP | 55.1±1.3 | 59.1±0.5 | 71.4±0.4 | 92.3±0.7 | 91.8±3.1 | 91.3±0.7 | 55.0±0.3 |
| GCN | 81.5±0.5 | 70.9±0.5 | 79.0±0.3 | 75.7±1.0 | 66.7±1.4 | 66.5±13.8 | 72.7±0.3 |
| GAT | 83.0±0.7 | 72.0±0.7 | 78.5±0.3 | 78.8±0.9 | 71.0±4.6 | 76.0±1.0 | 72.0±0.5 |
| GIN | 78.6±1.2 | 71.4±1.1 | 76.9±0.6 | 79.6±0.8 | 72.9±2.5 | 78.4±1.4 | 67.5±2.5 |
| GPRGNN | 83.8±0.9 | 75.9±1.2 | 79.8±0.8 | 75.9±6.2 | 89.9±3.0 | 85.0±5.2 | 70.4±1.5 |
| APPNP | 83.5±0.7 | **75.9±0.6** | 79.0±0.3 | 83.9±0.7 | 90.1±3.5 | 89.8±0.6 | 70.3±2.5 |
| H2GCN | 83.4±0.5 | 73.1±0.4 | 79.2±0.3 | 85.9±4.6 | 87.9±4.2 | 85.1±6.4 | 72.8±2.4 |
| SAGE | 74.5±0.8 | 67.2±1.0 | 76.8±0.6 | 79.3±1.2 | 64.8±5.2 | 71.4±1.2 | 70.6±1.6 |
| GRAND++ | 82.9±1.4 | 70.8±1.1 | 79.2±1.5 | 81.4±3.5 | 88.6±2.1 | 75.6±3.2 | 74.1±2.3 |
| ACMP | 82.2±1.5 | 71.4±0.5 | 79.1±2.3 | 89.2±5.5 | 88.1±4.2 | 85.6±4.5 | 72.1±0.4 |
| ChebNet | 81.2±0.7 | 69.8±1.5 | 74.4±0.3 | 83.5±2.6 | 86.1±3.5 | 80.4±1.7 | 70.8±0.4 |
| Framelets | 83.3±0.5 | 71.0±0.6 | 79.4±0.4 | 82.3±2.5 | 88.9±3.2 | 72.6±0.3 | 71.9±0.2 |
| DMD-GCN | 82.6±0.7 | 71.4±2.6 | 79.3±0.9 | 84.2±2.6 | 85.2±2.1 | 83.6±3.1 | 73.9±2.8 |
| DMD-SGC | **84.1±0.4** | 72.6±0.8 | 79.4±1.4 | 81.2±2.5 | 83.0±1.8 | 80.0±1.9 | 74.1±0.9 |
| DMD++ | 82.3±0.3 | 73.2±0.4 | 79.9±0.7 | **92.6±3.4** | **91.9±2.6** | **91.4±1.7** | 74.4±1.5 |
| DMD-ACMP | 82.9±0.9 | 73.0±2.1 | **81.2±1.5** | 89.4±3.8 | 89.4±2.2 | 88.1±2.4 | **75.5±0.8** |

## 7.1 NODE CLASSIFICATION ON HOMO AND HETEROPHILIC GRAPHS

We report the performances of DMD-GNNs via both homophilic and heterophilic graphs for later the connected nodes are often with distinguished labels. The baseline models include: two-layer MLP, GCN, GAT, GIN (Xu et al., 2019b),GPRGNN (Chien et al., 2021), APPNP (Gasteiger et al., 2019), H2GCN (Zhu et al., 2020), GraphSage (Hamilton et al., 2017), ACMP, GRAND++, ChebNet (Defferrard et al., 2016) and Graph Framelets (Zheng et al., 2021). For datasets, we include the homophilic datasets `Cora`, `Citeseer` and `Pubmed`, and heterophilic graphs `Texas`, `Wisconsin` and `Cornell`. In addition, to illustrate DMD-GNNs scalability, we test our models via `OGB-arXiv` (Hu et al., 2020).

For DMD-GNNs, we include **DMD-GCN** for which the initial feature dynamic is conducted via one layer of GCN without learnable weight matrices, i.e., $\mathbf{H}(\ell + 1) = \widehat{\mathbf{A}}\mathbf{H}(\ell)$. Followed by the generic form in equation (8), in terms of the initial non-linear dynamics, we selected the ACMP dynamic $\mathbf{H}(\ell + 1) = (\mathbf{A} - \mathbf{B})\mathbf{H}(\ell) + \mathbf{H}(\ell) \odot (\mathbf{I} - \mathbf{H}(\ell) \odot \mathbf{H}(\ell))$, leading to the **DMD-ACMP** model. We also define **DMD-SGC** with the initial dynamic as $\mathbf{H}(\ell + 1) = \widehat{\mathbf{A}}^s\mathbf{H}(\ell)$, where $s$ stands for a certain power of the adjacency and we fix $s = 2$ followed by the settings in SGC (Wu et al., 2019). Moreover, to let the initial dynamic appropriately capture the graph characteristics, we define the **DMD++** as $\mathbf{H}(\ell + 1) = \alpha\widehat{\mathbf{A}}^2\mathbf{H}(\ell) + (1 - \alpha)\mathbf{H}(\ell)$, suggesting a feature dynamic with two-hops of neighboring information (i.e., SGC) and source term. Additionally, we fixed the truncation rate $\xi = 0.85$ for homophilic graphs (as well as for `Ogbn-arXiv`) and $\xi = 0.7$ for heterophilic graphs.

The results are presented in Table 1, and one can check that DMD-GNNs demonstrate comparable performance to baseline models, with modest improvements on some datasets. In addition, DMD-SGC shows higher/lower accuracy compared to DMD-GCN via homophilic/heterophilic graphs, suggesting that a stronger smoothing/sharpening effect, i.e., $\widehat{\mathbf{A}}\mathbf{H}$ compared to $\widehat{\mathbf{A}}^2\mathbf{H}$ is preferred in homophilic/heterophilic graphs. Moreover, DMD-GNNs that are induced from the original dynamics that contain diffusion-reaction process (e.g., ACMP) show better performances than those DMD-GNNs induced from relatively simple dynamics (e.g., GCN), this implies that DMD-GNNs can inherit the advantages of their original models. Finally, one can observe that DMD-GNNs also outperform spectral GNNs such as ChebNet, suggesting that the produced from data-driven DMD (i.e., $\mathbf{\Psi}$ in equation (14)) effectively enhances GNN dynamics by offering a refined spectral domain.

## 7.2 PERFORMANCE ON LONG RANGE GRAPH BENCHMARKS

As DMD estimates the lower-rank graph adjacency matrix via a data-driven manner, thus, the resulting matrix $\mathbf{K}$ from equation (12) is, in general, denser than the original graph adjacency, leading the DMD-GNNs to own the natural advantage in terms of long-range graph learning tasks for which traditional GNNs tend to lose their accuracy due to reduced sensitivity between node features caused by the relatively long hops or distances between them, also known as the over-squashing issue (Shi et al., 2023b; Topping et al., 2021). In this experiment, we test the performance of DMD via two long-

Table 2: Results on **long-range graph datasets**, accuracies are reported via Macro-F1 scores. Best performances are highlighted in **bold**.

| Methods | COCO-SP | PascalVOC-SP |
|---------|---------|--------------|
| MLP | 0.031±0.016 | 0.114±0.023 |
| GCN | 0.079±0.025 | 0.238±0.016 |
| GPRGNN | 0.044±0.015 | 0.152±0.024 |
| SGC | 0.056±0.011 | 0.216±0.039 |
| DMD-GCN | 0.081±0.015 | **0.243±0.014** |
| DMD-SGC | 0.083±0.012 | 0.217±0.036 |
| DMD++ | **0.091±0.009** | 0.241±0.017 |

Table 3: Results on the **link prediction** tasks by treating DMD-GNNs as an graph feature encoder. The best performances are highlighted in **bold**.

| Methods | Cora | Citeseer | Chameleon | Squirrel |
|---------|------|----------|-----------|----------|
| MLP | 75.0±1.1 | 78.3±0.8 | 76.9±0.4 | 73.2±0.2 |
| GCN | 76.9±0.8 | 77.9±0.9 | 78.7±0.5 | 74.7±0.1 |
| SAGE | 75.1±0.5 | 76.3±1.2 | **82.1±0.4** | 75.3±0.3 |
| APPNP | 76.0±0.9 | 75.9±1.1 | 78.9±0.8 | 73.6±0.2 |
| DMD-GCN | 75.7±0.4 | **78.7±0.6** | 76.1±0.4 | **78.8±0.4** |
| DMD-SGC | **77.4±0.2** | 77.2±0.7 | 78.3±0.7 | 74.3±1.2 |
| DMD++ | 75.3±0.6 | 76.6±0.6 | 77.5±0.3 | 78.1±0.5 |

Table 4: Results (MSE) on spatial-temporal prediction datasets, with the best highlighted in **bold**.

| Dataset | MLP | GCN | GAT | GAT-KNN | GCN-KNN | ChebNet | Grand++ | Framelet | DMD-GCN | DMD-SGC | DMD-ACMP | DMD++ |
|---------|-----|-----|-----|---------|---------|---------|---------|----------|---------|---------|----------|-------|
| Chickenpox | 0.924 (±0.001) | 0.923 (±0.001) | 0.924 (±0.002) | 0.926 (±0.004) | 0.926 (±0.004) | 0.916 (±0.014) | 0.991 (±0.009) | 0.923 (±0.03) | 0.978 (±0.093) | 0.935 (±0.014) | 0.924 (±0.001) | **0.910** (**±0.002**) |
| Covid | 0.956 (±0.034) | 0.956 (±0.029) | 1.052 (±0.081) | 0.861 (±0.045) | 1.475 (±0.028) | 0.991 (±0.078) | 1.271 (±0.073) | 0.938 (±0.081) | **0.760** (**±0.025**) | 0.805 (±0.067) | 0.809 (±0.011) | 0.784 (±0.013) |
| WikiMath | 1.073 (±0.042) | 1.292 (±0.125) | 1.339 (±0.073) | **0.826** (**±0.070**) | 1.023 (±0.058) | 1.589 (±0.048) | 1.044 (±0.328) | 0.988 (±0.069) | 0.848 (±0.031) | 0.901 (±0.028) | 0.934 (±0.015) | 0.805 (±0.042) |

range graph datasets, namely `PascalVOC-SP` and `COCO-SP` (details in Appendix C.3). We did not include DMD-ACMP in this experiment due to the potential high variation of its hyperparameters. We evaluate the model performance via Macro-F1 scores and the final results are averaged from 10 runs. The learning outcomes are presented in Table. 2. One can see DMD-GNNs show superior performances by surpassing baseline models via `COCO-SP`, whereas limited improvements can be observed via `PascalVOC-SP`. We noticed that this might be due to the fact that in both datasets, the initial feature dimension is less than the number of classes, and DMD-GNNs are conducted via the spectral filtering paradigm. Hence, limited free parameters can be adjusted. In Appendix C.3.4, we show how we slightly changed our model's architecture to better handle this phenomenon.

## 7.3 SPATIAL-TEMPORAL DYNAMICS PREDICTION FOR INFECTIOUS DISEASE EPIDEMIOLOGY

We test DMD-GNNs in three spatial-temporal datasets. These datasets are a series of graph-structured data with each of them containing an integer observation, e.g., the Chickenpox and COVID prevalence. The task is to predict the future, e.g., disease occurrence based on the historical records (snapshots). For example, in the `Chickenpox` dataset, the initial graph is with the nodes as countries and edges as direct neighborhood relationships. Followed by the settings from (Wu et al., 2023a), the node features are lagged weekly counts of the chickenpox cases (e.g., 4-lags). For a more appropriate comparison, we also deploy GCN and GAT via KNN graphs. However, we do not use the K-Nearest-Neighbors (KNN) graph for DMD-GNNs. Table 4 shows the prediction accuracy of DMD-GNNs compared to different competitors. Other than GCN and GAT models, we also include ChebNet, Grand++, Graph Framelets DMD-GNNs show remarkable performances among all included datasets, indicating the potential of deploying DMD and related approaches to identifying the principal spreading pattern in terms of infectious disease epidemiology, see Appendix C.4.2 for more details.

## 7.4 SENSITIVITY ANALYSIS

In this experiment, we test DMD-GNN's sensitivities on the parameter $\xi \in [0.2, 0.8]$, which controls the rate of truncation in DMD. The results are shown in Figure 2(a). One can observe that in homophilic graphs, the best results for DMD-GNNs are achieved in a relatively large value of $\xi$ (i.e., using a wider spectrum) compared to heterophilic graphs in which learning accuracy dropped after best performances with lesser $\xi$ are achieved. This aligns with our previous claim in Section 6. That is, the graph structural information, as well as its spectrum, is with higher/lower importance via homophilic/heterophilic graphs.

## 7.5 DMD-GNNS FOR LINK PREDICTION

In this section, we demonstrate that DMD-GNNs can serve as a powerful encoder for link prediction tasks by effectively capturing complex relational structures in graphs. Based on the work in (Kipf & Welling, 2016), we design a model in which during the encoding process, negative links are randomly

Figure 2: (a): hyperparameter sensitivity. (b): comparison between DMD-GNNs and PIDMD-GNNs in directed and undirected graphs.

added to the original graph to enhance the model's discriminative capabilities. The model is then tasked with a binary classification problem, distinguishing between positive links from the original edges and negative links from the artificially added edges. Using node embeddings generated by the DMD-GNN encoder, the decoder predicts link existence across all edges including the negative links by computing the dot product between pairs of node embeddings for each edge. This operation aggregates the values across the embedding dimensions to produce a single scalar that represents the probability of edge existence for each edge. We select `Cora`, `Citeseer`, `Chameleon` and `Squirrel` (Rozemberczki et al., 2021a) for the testing datasets and fixed the quantity of $\xi$ same as the ones used in node classification. The results are presented in Table. 3. We note that we use accuracy as the metric instead of AUC-ROC because we adopt equal-proportion sampling for positive and negative edges. One interesting observation here is the DMD-GNNs included from a relatively less complex initial dynamic (e.g., GCN) show better link prediction accuracy compared to others, and MLP shows comparable learning outcomes compared to all GNN models.

## 8 MODEL EXTENSIONS AND FURTHER DIRECTIONS

**DMD with Additional Constraints** One can check that DMD approach defined in Section 5.1 can also be viewed as (Schmid, 2010; Tu, 2013) $\arg\min_{\mathrm{rank}(\mathbf{K}) \leq d} \|\mathbf{H}(\ell+1) - \mathbf{K}\mathbf{H}(\ell)\|_F$. However, when information about the underlying dynamic is known, e.g., $\mathbf{K}$ should be symmetric or shift equivariant, one may prefer to align these constraints to the above optimization problem, yielding the so-called physics-informed DMD (Baddoo et al., 2023) as $\arg\min_{\mathbf{K} \in \mathcal{M}} \|\mathbf{H}(\ell+1) - \mathbf{K}\mathbf{H}(\ell)\|_F$, where we let $\mathcal{M}$ be the generic notion of the potential manifold that contains $\mathbf{K}$. The manifold constraint on $\mathbf{K}$ leads to variations in terms of the estimated eigenvectors, i.e., $\mathbf{\Psi}$, indicating a family of physics-informed DMD-GNNs (PIDMD-GNNs), that can potentially show better performances in different learning tasks. In addition, one can let $\mathbf{K}$ satisfy the so-called coupling conditions (Shi et al., 2021), such that $\mathbf{K}\mathbf{1} = \mu, \mathbf{K}^\top \mathbf{1} = \nu$, then the problem can be treated as an optimal transport problem between features, and these features may not need to be sourced from the same graph, e.g., transportation between two knowledge graphs. In this case, $\mathbf{K}$ serves as a graph-structured transportation plan between two distributions.

**On Directed Graphs and Reconstruct Complex Physical Systems** The DMD algorithm described in Section 5.1 is not necessarily produce a symmetric $\mathbf{K}$. This suggests DMD algorithm may be naturally suitable for direct graphs. To verify this idea and motivate future research directions, we briefly test the performances of DMD-GNNs and PIDMD-GNNs (with symmetric constraints) via two directed graphs: `Computer` and `Photo` in which, unlike the node classification experiment where the graph is programmed to be undirected, we preserve the directions of the links. The results presented in Figure. 2(b) show that PIDMD-GNNs underperform/outperform DMD-GNNs via directed/undirected graphs. In Appendix C.6 we also show that PIDMD-GNNs own the potential to reconstruct complex physical dynamics such as Schrödinger's equation.

## 9 CONCLUSION

In this work, we introduced a novel integration of DMD with GNNs, demonstrating its efficacy in enhancing GNN performance across various graph learning tasks. By leveraging DMD to approximate the underlying dynamics of graph data, we are able to capture key dynamical patterns through a low-rank representation, which in turn enables more effective feature propagation in GNNs. Our proposed DMD-GNN models demonstrate superior performance across various tasks, underscoring the benefits of incorporating the advanced dynamic system analysis tool into graph learning.

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

APPENDICES CONTENTS

## A THEORETICAL JUSTIFICATIONS OF DMDs

### A.1 THEORETICAL PROOFS

**Lemma (Formal Version of Lemma 1).** *Assuming that the system defined in equation (15) has a stable hyperbolic fixed point at origin $\mathbf{x} = 0$, $\mathbf{f} \in \mathcal{C}^2$ in a neighborhood of the origin. Assuming the measurable function $\boldsymbol{\Phi} = \mathrm{id}$, i.e., $\boldsymbol{\Phi}(\mathbf{X}) = \mathbf{X}$, and for some (generic) integer d, $\mathrm{rank}\left[D\mathbf{X}(\ell)|_S\right] = d$, suggesting no rank degeneracy in the slow decayed subspace S. Further assume that $\mathrm{rank}(\mathbf{H}) = d$ and $|\mathbf{U}(\mathcal{A})_F\mathbf{X}(\ell)|, |\mathbf{U}(\mathcal{A})_F\mathbf{X}(\ell + 1)| \leq |\mathbf{U}(\mathcal{A})_S\mathbf{X}(\ell)|^{1+\tau}$ for some $\tau \in (0, 1]$, suggesting in the underlying dynamics the subspace outside S can quickly shrink out. Then the estimation of DMD, denoted as $\mathcal{D}$ (e.g., $\mathbf{K} \subseteq \mathcal{D}$) is with the form*

$$\mathcal{D} = \mathbf{U}(\mathcal{A})_S \mathrm{e}^{\boldsymbol{\Lambda}_S \Delta t} \mathbf{U}(\mathcal{A})_S^\top + \mathcal{O}(|\mathbf{U}(\mathcal{A})_S\mathbf{X}(\ell)|^\tau), \tag{18}$$

*and $\mathcal{D}$ is locally topologically conjugated with order $\mathcal{O}(|\mathbf{U}(\mathcal{A})_S\mathbf{X}(\ell)|^\tau)$ error to the linearized dynamic on a d-dimensional, slow attracting spectral submanifold $\mathcal{M}(S) \in \mathcal{C}^1$ tangent to S at $\mathbf{x} = 0$.*

Our proof follows directly from the one used in proving Theorem 1 in the work (Haller & Kaszás, 2024), with the case that $\mathcal{A}$ is symmetric. We briefly show our proof of self-completeness.

*Proof.* By the assumption of a stable hyperbolic fixed point and the $\mathcal{C}^1$ linearization theorem (Hartman, 1960), any trajectory in a neighborhood of the origin in the nonlinear system equation (15) converges at an exponential rate $\mathrm{e}^{\lambda_{d+1}t}$ to a d-dimensional attracting spectral submanifold $\mathcal{M}(\mathcal{S})$, tangent to a d-dimensional attracting slow spectral subspace S of the linearized system at the origin. Based on Theorem 1 in (Haller & Kaszás, 2024), we have the form of the estimations of DMD as the submanifold defined as

$$\mathcal{D} = D\boldsymbol{\Phi}(\mathbf{0})\mathbf{T}(\mathcal{A})_S \mathrm{e}^{\boldsymbol{\Lambda}_S \Delta t}(D\boldsymbol{\Phi}(\mathbf{0})\mathbf{T}(\mathcal{A})_S)^{-1} + \mathcal{O}(|\mathbf{P}(\mathcal{A})_S\mathbf{X}(\ell)|^\tau), \tag{19}$$

where we let $\mathbf{T}(\mathcal{A})_S$ and $\mathbf{P}(\mathcal{A})_S \in \mathbb{R}^{N \times d}$ be the right and left eigenvectors of $\mathcal{A}$ restricted by S, respectively. Given we assumed the $\mathcal{A}$ is symmetric, we have $\mathbf{T}(\mathcal{A})_S = \mathbf{P}(\mathcal{A})_S = \mathbf{U}(\mathcal{A})_S$. Therefore the above equation can be further simplified to

$$\mathcal{D} = D\boldsymbol{\Phi}(\mathbf{0})\mathbf{U}(\mathcal{A})_S \mathrm{e}^{\boldsymbol{\Lambda}_S \Delta t}(D\boldsymbol{\Phi}(\mathbf{0})\mathbf{U}(\mathcal{A})_S)^{-1} + \mathcal{O}(|\mathbf{U}(\mathcal{A})_S\mathbf{X}(\ell)|^\tau). \tag{20}$$

Further, because the assumption that $D\boldsymbol{\Phi} = \mathrm{id}$, hence the derivative $\mathcal{D} = D\boldsymbol{\Phi}(\mathbf{X}(\ell))$ w.r.t $\mathbf{X}(\ell)$ is $\mathbf{I}$, therefore we have

$$\mathcal{D} = \mathbf{U}(\mathcal{A})_S \mathrm{e}^{\boldsymbol{\Lambda}_S \Delta t} \mathbf{U}(\mathcal{A})_S^\top + \mathcal{O}(|\mathbf{U}(\mathcal{A})_S\mathbf{X}(\ell)|^\tau), \tag{21}$$

and this completes the proof. □

## B GNNs WITH DIFFERENT TYPES OF SOURCE TERMS

In this section, we provide more examples regarding GNNs with source terms, which serve as a **generic** notion to those GNNs that contain two terms with one term homogenizing the node feature and another term bringing the variation back to the system to preserve the dissimilarities between features. In terms of the spatial GNNs, this paradigm typically refers to the diffusion-reaction paradigm mentioned in the paper. Based on the summary provided in Choi et al. (2023) and Han et al. (2024), we include some of recently developed GNNs for self-completeness. On the other hand, in terms of spectral GNNs, this GNN structure may refer to multi-scale GNNs like Wavelet GNNs (Xu et al., 2019a) or Framelet GNNs (Shi et al., 2023a; Zheng et al., 2022; Shao et al., 2023; Shi et al., 2023c). Given there are many GNNs in this field, in this work, we only include the formulation of the models that we included in our experiments, for more details, please refer to the recent reports (Han et al., 2024). In terms of the notations, for all formulated GNNs, we denote them via normalized adjacency/Laplacian matrices, i.e., $\widehat{\mathbf{A}}$ and $\widehat{\mathbf{L}}$, respectively, although notation could be a bit different within their original publications. We further note that we omit to introduce those basic yet state-of-the-art GNNs such as MLP, GCN, GAT and GPRGNN. Finally, we start by showing the formulation of the APPNP model which has the feature propagation as:

$$\text{APPNP}: \mathbf{H}(\ell + 1) = (1 - \alpha)\widehat{\mathbf{A}}\mathbf{H}(\ell) + \alpha\mathbf{H}(0), \quad \mathbf{H}(\text{out}) = \mathbf{H}(L)\mathbf{W}, \tag{22}$$

where we let $L$ be the number of layers. The core idea of APPNP is to balance the diffusion and reaction terms by $\alpha$ and $1 - \alpha$, and this approach has been adopted by many successive models, such as ACMP (Wang et al., 2023):

$$\mathbf{H}(\ell + 1) = (\widehat{\mathbf{A}} - \mathbf{B})\mathbf{H}(\ell) + \mathbf{H}(\ell) \odot (\mathbf{I} - \mathbf{H}(\ell) \odot \mathbf{H}(\ell)) \tag{23}$$

in which, a matrix $\mathbf{B}$ is involved with adjacency matrix $\widehat{\mathbf{A}}$ for adjusting the attractive and repulsive forces between nodes, and the so-called Allen-Cahn term $\mathbf{H}(\ell) \odot (\mathbf{I} - \mathbf{H}(\ell) \odot \mathbf{H}(\ell))$ is leveraged as the source term. Similar to APPNP, the graph sample and aggregate (SAGE) (Hamilton et al., 2017) is with the propagation rule as:

$$\text{SAGE} : \mathbf{H}(\ell + 1) = \mathbf{H}(\ell)\mathbf{W}_1(\ell) + \widehat{\mathbf{A}}\mathbf{H}(\ell)\mathbf{W}_2(\ell), \tag{24}$$

where the source term is generated from the (embedded) feature matrix in addition to the original GCN propagation. We then formulate the GRAND++, which serves as the initial work on bringing the source term to the diffusion process of the graph $\frac{\partial \mathbf{H}}{\partial t} = \text{div}(\widehat{\mathbf{A}}(\mathbf{H}) \cdot \nabla \mathbf{H}) + \mathbf{S}$ which owns discretized version as:

$$\text{GRAND++} : \quad \mathbf{H}(\ell + 1) = \widehat{\mathbf{A}}\mathbf{H}(\ell) + \mathbf{S}, \tag{25}$$

in which the $\mathbf{S}$ is the source term with each row of the $\mathbf{S}$ is defined as $\mathbf{S}_i = \sum_{j \in \mathcal{I}} \delta_{ij}(\mathbf{H}(0)_j - \bar{\mathbf{H}}(0)$ with $\mathcal{I} \subseteq \mathcal{V}$ a selected node subset used as the source term and $\bar{\mathbf{H}}(0) = \frac{1}{|\mathcal{I}|}\sum_{j \in \mathcal{I}} \mathbf{H}(0)_j$ is the average signal. $\delta_{ij}$ denotes the initial transition probability from node $j$ to $i$. Finally, we include H2GCN as

$$\text{H2GCN} : \mathbf{H}_{i,h}(\ell + 1) = \text{AGGR}\{\mathbf{H}_j(\ell) : j \in \mathcal{N}_h(i)\} = \sum_{j \in \mathcal{N}_h(i)} \mathbf{H}_i(\ell)d_{i,h}^{1/2}d_{j,h}^{1/2}, \text{and} \tag{26}$$

$$\mathbf{H}_i^{\text{final}} = \text{Combine}(\mathbf{H}_i(0), \mathbf{H}_i(1), \cdots \mathbf{H}_i(L)), \tag{27}$$

where we let $h$ be the neighboring order (hops) that H2GCN uses to aggregate, and the representation is formed as a combination of all intermediate representations. Lastly, another embedding of $\mathbf{H}^{\text{final}}$ is leveraged to make the final prediction of the labels.

In terms of spectral GNNs, we only include ChebNet (Defferrard et al., 2016) and graph framelets. Both two models propagate the node features via spectral filtering approach. For example, we have

$$\text{ChebyNet} : \mathbf{H}(\ell + 1) = \mathbf{U}\text{diag}(\boldsymbol{\theta})\mathbf{U}^\top\mathbf{H}(\ell), \tag{28}$$

in which we let $\text{diag}(\theta)$ be the filtered graph spectrum. We note that the process of the ChebyNet can be empirically approximated by the corresponding polynomials and one usually applies feature embedding, e.g., $\text{MLP}(\mathbf{H})$, prior to the spectral filtering process for computational convenience. Similarly, framelet GNN attempts to balance the feature smoothing and sharpening effects induced from the spectral filtering on both low and high pass domains, that is

$$\text{Framelet} \quad \mathbf{H}(\ell + 1) = \sum_{(r,j) \in \mathcal{P}} \mathcal{W}_{r,j}^\top \text{diag}(\boldsymbol{\theta}_{r,j})\mathcal{W}_{r,j}\mathbf{H}(\ell)\mathbf{W}(\ell), \tag{29}$$

where we let $\mathcal{W}$ be the framelet decomposition matrices such that $\sum_{(r,j) \in \mathcal{I}} \mathcal{W}_{r,j}^\top \mathcal{W}_{r,j} = \mathbf{I}$ for $\mathcal{P} = \{(r,j) : r = 1, ..., L, j = 0, 1, ..., J\} \cup \{(0, J)\}$. For example, one can denote the framelet Haar type filter as

$$\mathbf{H}(\ell + 1) = \mathbf{U}\cos^2(\text{diag}(\boldsymbol{\theta}))\mathbf{U}^\top\mathbf{H}(\ell)\mathbf{W}(\ell) + \mathbf{U}\sin^2(\text{diag}(\boldsymbol{\theta}))\mathbf{U}^\top\mathbf{H}(\ell)\mathbf{W}(\ell), \tag{30}$$

and let $\boldsymbol{\theta}_{ii} > 0$, one can check that compared to the low pass filtering process $\cos(\text{diag}(\boldsymbol{\theta}))\mathbf{U}^\top\mathbf{H}(\ell)\mathbf{W}(\ell)$, one can treat the high pass filtering process as a reaction process to bring the variation back to the model. Finally, along with the development of the GNNs for the node-level classification task, a number of GNNs are proposed motivated by maximizing their expressive power (Xu et al., 2019b; Azizian & marc lelarge, 2021; Morris et al., 2023) targeting on the graph-level classification tasks (i.e., graph pooling). In general, a powerful GNN shall be equivalent to the so-called Weisfeiler and Leman (WL) test (Leman & Weisfeiler, 1968) to distinguish any two non-isomorphic graphs. Accordingly, in our experiment, we also include one of the fundamental GNNs with the maximal expressive power, graph isomorphism networks (GIN), in which, before the final pooling layer, the propagation of the node features can be denoted as

$$\mathbf{H}(\ell + 1) = \text{MLP}(\ell)\left((1 + \epsilon(\ell))\mathbf{H}(\ell) + \mathbf{A}\mathbf{H}(\ell)\right), \tag{31}$$

where we let $\epsilon$ be the scaling parameter added to self-loops after diagonalization.

## B.1 Linear and Non-linear GNN Dynamics

In addition to our statement in Section 4, here we provide more discussion on the linear/non-linearity of the GNN dynamics. The main motivation for designing non-linear GNNs is to enhance GNN dynamics to fit more complex graph datasets and downstream tasks. For example, it is common to observe that in spectral GNNs, the spectral filtering functions are learned via non-linear domains, and higher-order polynomials are usually leveraged to approximate them (Defferrard et al., 2016; Zheng et al., 2021). On the other hand, non-linearity dynamics are widely utilized in spatial GNNs as well in order to let models propagate richer feature information from wider receptive fields (e.g., SGC). Compared to linear GNNs, those non-linear GNNs usually have higher computational complexity and lesser interpretability to exchange the model accuracy. Driven by the snapshots induced by these complex dynamics, DMD offers us the potential to analyze and approximate these dynamics in a simple linear way, and our proposed DMD-GNNs can also achieve similar or even better empirical results compared to their non-linear GNN counterparts.

## C  Experiment Details

In this section, we first present the pseudocode of our DMD-GNNs, followed by the details of our experiments, and in the next section, we present some extra tests in addition to our current results. Our code is available at `https://github.com/EEthanShi/Graph-DMD`.

### C.1  Pseudocode of DMD-GNN

In addition to the flow chart in Figure. 1, below we show the pseudocode of DMD algorithm and training of DMD-GNN.

---

**Algorithm 1** Training Algorithm for DMD-GNN (Classification)

---

**Input:** Input Graph adjacency $\mathbf{A}$, initial GNN model and ground truth $\mathbf{Y}$.
**Output:** DMD-GNN prediction Accuracy
1: Apply the initial GNN model (e.g., GCN) to obtain system snapshots (i.e., $\mathbf{H}(\ell + 1)$, $\mathbf{H}(\ell)$) through $\mathbf{H}(\ell + 1) = \mathbf{A}\mathbf{H}(\ell)$.
2: Apply DMD in Algorithm 2 to obtain DMD modes $\boldsymbol{\Psi}$.
3: **while** Not converged **do**
4:    Spectral filtering via the domain defined by $\boldsymbol{\Psi}$, $\mathbf{H}(\ell + 1) = \boldsymbol{\Psi}\mathrm{diag}(\boldsymbol{\theta})\boldsymbol{\Psi}^\top \mathbf{H}(\ell)\mathbf{W}(\ell)$,
5:    Compute the model output $\widehat{\mathbf{Y}} = \mathrm{Softmax}(\mathbf{H}(\ell + 1))$,
6:    Take the gradient step on $\boldsymbol{\theta}$, $\mathbf{W}$

$$\nabla_{\boldsymbol{\theta},\mathbf{W}}\mathrm{CrossEntropy}(\widehat{\mathbf{Y}}).$$

7: **end while**

---

**Algorithm 2** Dynamic Mode Decomposition (DMD) Algorithm

---

**Input:** Snapshots of the system: $\mathbf{H}(\ell)$ and $\mathbf{H}(\ell + 1)$, truncation rate $\xi$.
**Output:** DMD modes $\boldsymbol{\Psi}$, *Optional:* Reconstructed operator $\mathbf{K}$.
1: Perform singular value decomposition (SVD) of $\mathbf{H}(\ell)$, $\mathbf{H}(\ell) = \mathbf{M}\boldsymbol{\Sigma}\mathbf{V}^*$.
2: Truncate the SVD results based on the truncation rate $\xi$ to rank $r$: $\boldsymbol{\Sigma}_r$, $\mathbf{M}_r$, $\mathbf{V}_r$.
3: Compute the reduced operator: $\mathbf{K} = \mathbf{M}_r^* \mathbf{H}(\ell + 1)\mathbf{V}_r\boldsymbol{\Sigma}_r^{-1}$.
4: Compute eigendecomposition of $\mathbf{K}$: $\mathbf{K}\mathbf{U}(\mathbf{K}) = \mathbf{U}(\mathbf{K})\boldsymbol{\Lambda}(\mathbf{K})$.
5: Compute the DMD modes: $\boldsymbol{\Psi} = \mathbf{H}(\ell)\mathbf{V}\boldsymbol{\Sigma}^{-1}\mathbf{U}(\mathbf{K})$.

---

We note that $\Sigma_r$, $\mathbf{M}_r$ and $\mathbf{V}_r$ are with reduced dimension (e.g., $r$) due to the truncation.

DMD-GNN has three major components in terms of computation: GNN, DMD and spectral filtering. Apparently, DMD-GNN scales with the chosen GNN. DMD is a one time computation and essentially an SVD where efficient method such as (Lim et al., 2015) can be applied. The spectral filtering in

Table 5: Summary statistics of datasets, the column $\mathbf{H}(\mathcal{G})$ is the homophily index.

| Datasets | #Classes | #Features | #nodes | #Edges | $\mathbf{H}(\mathcal{G})$ | Ranks |
|---|---|---|---|---|---|---|
| Cora | 7 | 1433 | 2708 | 5429 | 0.825 | 421 |
| Citeseer | 6 | 3703 | 3327 | 4372 | 0.717 | 819 |
| Pubmed | 3 | 500 | 19717 | 44338 | 0.792 | 295 |
| CS | 15 | 6805 | 18333 | 100227 | 0.832 | 1750 |
| Photo | 8 | 745 | 7487 | 126530 | 0.849 | 402 |
| Ogbn-arxiv | 47 | 100 | 169343 | 1166243 | 0.681 | 128 |
| Wisconsin | 5 | 251 | 499 | 1703 | 0.150 | 21 |
| Texas | 5 | 1703 | 183 | 279 | 0.097 | 35 |
| Cornell | 5 | 1703 | 183 | 277 | 0.386 | 16 |

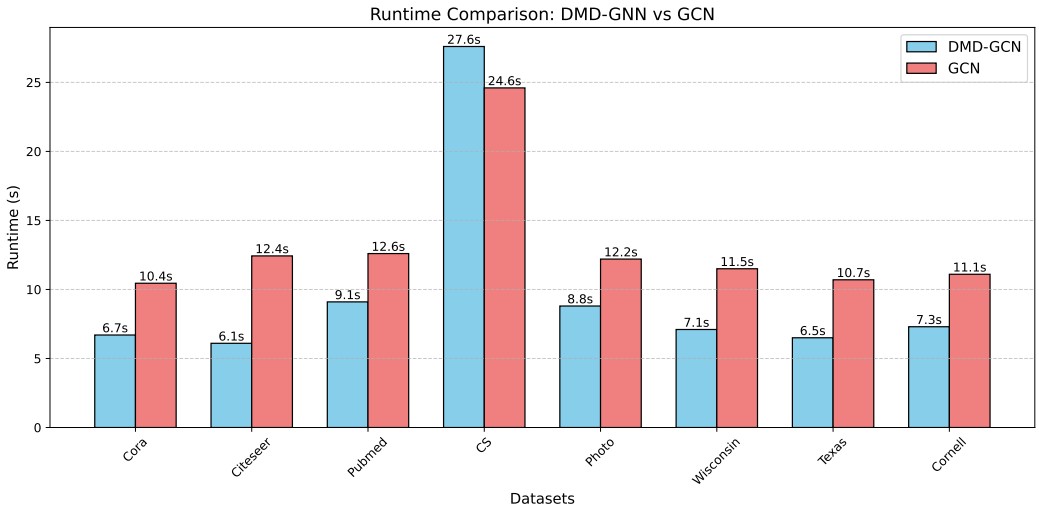

Figure 3: Comparison between GCN (two-layers) and DMD-GCN of the total run times for citation datasets.

Line 4 in Algorithm 1 has similar scalability as a spectral GNN, although $\boldsymbol{\theta} \in \mathbb{R}^r$ and $r < \min(N, d)$ thanks to the low rank in DMD. Therefore, the scalability of DMD-GNN is mainly determined by the GNN used and a spectral GNN.

## C.2 NODE CLASSIFICATION ON BOTH TYPES OF GRAPHS

Table. 5 shows the summary statistics of the included static graphs, in which we let $\mathbf{H}(\mathcal{G})$ be their homophily index which indicates the uniformity of labels in neighborhoods. For example, a higher homophily index suggests that the connected nodes within the graph are more likely with the same label. we note that we also show the summary statistics of `CS` and `Photo` as they have been utilized via Section 8 for testing the performance difference between DMD-GNNs and PIDMD-GNNs. Finally, as we have shown in the main page, we fixed SVD Rank within the SVD of $\mathbf{H}(\ell)$ as 0.85 for the homophilic graphs (as well as for `Ogbn-arXiv`)) and 0.7 for heterophilic graphs. In Table. 5 we also show the number of actual ranks that are used for DMD-GNNs, and these ranks directly determine the number of parameters for the spectral filtering within DMD-GNNs, e.g., equation (14). Figure. 4 visualizes the actual number of the ranks (highlighted in orange) via six included datasets. From both table and figure, one can see that DMD significantly reduced the number of singular values. Especially in the heterophilic graphs, DMD-GNNs only require less than 40 learnable parameters for spectral filtering. This directly verified our claim in Section 5.2. That is, for homophilic graphs, in which more graph structural information is needed for GNNs, one may need a higher value of $\xi$ for the truncation compared to the heterophilic graphs.

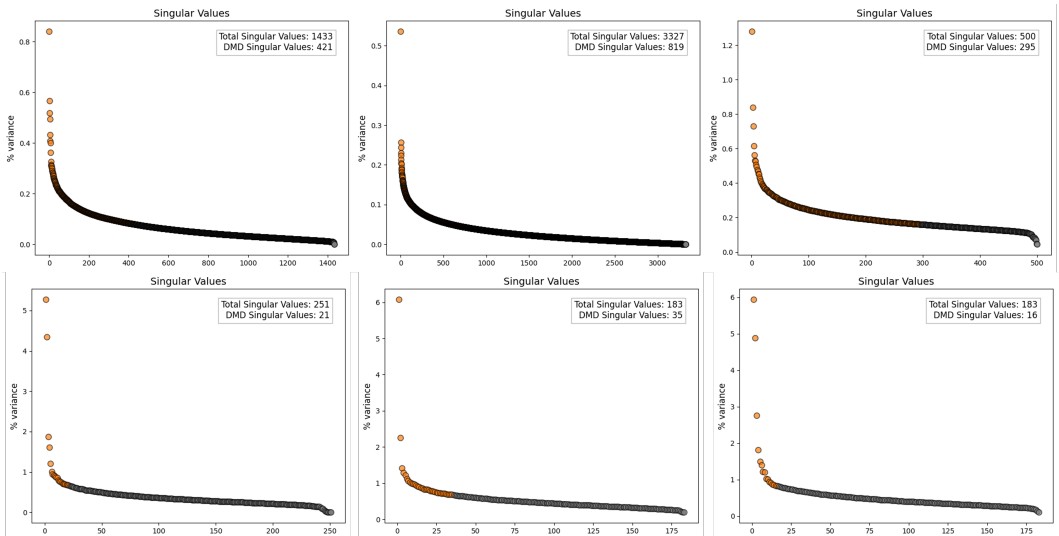

Figure 4: Number of feature dimensions and the actual number of ranks selected from the DMD. From the first row ($\xi = 0.85$): Cora, Citeseer, Pubmed; second row ($\xi = 0.7$): Wisconsin, Texas and Cornell.

### C.2.1 BASELINES AND DMD-GNNS IMPLEMENTATION DETAILS

All included baselines are implemented using torch.geometric, except graph framelets with we adapt from the work in (Zheng et al., 2022; Yang et al., 2024). As there are many ways of implementing graph framelets, in this work, we only consider the easiest case, i.e., Haar filter, which leads to the propagation denoted in equation (30). We set the number of layers of all included baselines as 2, and the baseline outcomes are collected from the pre-published results. In terms of DMD-GNNs, all of our proposed models adopt a spectral filtering paradigm, as illustrated in equation (14). Hence, in most of our conducted experiments, we let all DMD-GNNs with one single layer. More specifically, for all tested datasets, we do normalization on both feature and graph adjacency to adjust the potential scaling issues. In terms of the ACMP dynamic, we fixed $\mathbf{B}_{ij} = 0.0001$ and 0.5 for homophilic and heterophilic graphs, respectively based on the conclusion in (Wang et al., 2023)

For all included datasets, we followed the standard data split scheme. In terms of the model training and testing, we leverage the ADAM as the optimizer. We also found that in this experiment, our proposed DMD-GNN models are not sensitive to hyperparameters such as hidden dimension, learning rate, weight decay, dropout, etc, suggesting the effectiveness of incorporating DMD to refine the initial dynamics on the graph. Finally, the maximum number of epochs is set as 200 for all included datasets except 500 for Ogbn-arXiv. The average test accuracy and its standard deviation came from 10 runs. In Figure. 3, we show the total training time for DMD-GCN and GCN (two layers) via both homophilic and heterophilic graph datasets. We highlight that as our DMD-GNNs propagate node features via spectral filtering, hence we implemented DMD-GNNs usually with one layer, which aligns with the settings in Defferrard et al. (2016). In terms of the comparison between the number of parameters, here we take Cora dataset, with feature and label dimensions 1433 and 7, respectively, as an example, and we compare the parameters within GCN (two-layers), ChebNet (one-layer), DMD-GCN (one-layer). Let the number of hidden dimensions be 16. One can check two-layer GCN with the total parameters as $1433 \times 16 + 16 \times 7 + 16 + 7 = 23063$ (with bias), and for ChebNet with polynomial order of 2 is $2 \times 1433 \times 16 + 2 \times 16 \times 7 + 17 + 7 = 46103$. In terms of DMD-GNNs, as we adopt a spectral filtering paradigm, the number of parameters is $1433 \times 16 + 421 = 23349$, which is very similar to GCN two layers but far less than ChebNet. This directly suggests the effectiveness of incorporating DMD to the GNN dynamics.

Table 6: Summary statistics of the selected LRGBs (Dwivedi et al., 2022)

| Dataset | Total Graphs | Total Nodes | Average Nodes | Total Edges |
|---|---|---|---|---|
| **PascalVOC-SP** | 11355 | 5443545 | 479.40 | 30777444 |
| **COCO-SP** | 123286 | 58793216 | 476.8 | 332091902 |

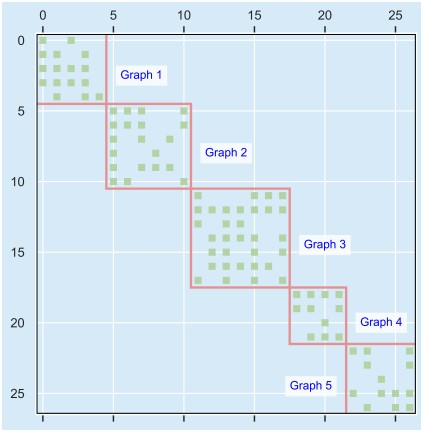

Figure 5: Illustration of the block-wise adjacency matrix used in LRGB experiment. The first edge of the second graph is with the index right after the first graph.

## C.3 PERFORMANCE ON LONG RANGE GRAPH BENCHMARKS (LRGB)

### C.3.1 OVER-SQUASHING IN GNNS

In this section, we demonstrate the motivation for testing DMD-GNNs on long-range graph datasets. The over-squashing (OSQ) problem was initially identified in (Alon & Yahav, 2021) as a phenomenon, termed the *bottleneck*, that arises during message aggregation by graph nodes along long paths. The presence of this bottleneck leads to exponentially increasing information being compressed into fixed-size vectors as the number of layers in a Message Passing Neural Network (MPNN) increases that is, as the network becomes "deeper" (Topping et al., 2021). Consequently, graph-based MPNNs are unable to effectively transmit messages from distant nodes, resulting in poor performance when the prediction task depends on long-range interactions. Although the OSQ problem was initially identified empirically in (Alon & Yahav, 2021), subsequent work by Topping et al. (2021) first attempted to quantify the phenomenon by analyzing the sensitivity of each node representation $\mathbf{h}_i(\ell)$ with respect to the input $\mathbf{x}_s$, denoted by $\frac{\partial \mathbf{h}_i(\ell)}{\partial \mathbf{x}_s}$. We refer to the work done by Shi et al. (2023b) for a more detailed discussion on the over-squashing issue.

In general, there are two major ways of mitigating the over-squashing issue, namely spatial or spectral rewiring. Specifically, spatial methods conduct rewiring via the graph topological indicators such as curvatures (Topping et al., 2021; Black et al., 2023; Giraldo et al., 2023; Fesser & Weber, 2023) whereas spectral methods mainly target optimizing the graph Laplacian spectrum (Deac et al., 2022; Banerjee et al., 2022). We highlight there are a few methods that alleviate the over-squashing issue in an implicit manner (Shi et al., 2024b; Chamberlain et al., 2021a;b), however, we omit to discuss these methods in detail as this is out of the scope of our work. Nevertheless, long-range graph benchmarks, provided in (Dwivedi et al., 2022) have been leveraged in many subsequent works for testing the model's performance and mitigation power on the over-squashing issue.

### C.3.2 DATASETS AND DATA PRE-PROCESSING

Regarding the datasets, in this work, we select two datasets in LRGB from `Pytorch-geometric`, namely `COCO-SP` and `PascalVOC-SP`, both original designed for the computer vision tasks (Dwivedi et al., 2022). Specifically, given the input image, the image segmentation techniques (e.g., SLIC (Achanta et al., 2012)) are initially applied to split the image by the so-called superpixels (SP). These superpixels contain rich information such as coordinates and RGB components as

the representations of different areas of the image. Then based on the similarities between SPs, graph generation methods such as KNN are leveraged to build up the graph in the image. Each SP corresponds to a region of the image belonging to a particular class. We show the summary statistics of `COCO-SP` and `PascalVOC-SP` in Table. 6.

Given the large number of nodes and edges in both datasets, in our presented experiment, we only select 10% of total graphs for training our model. In addition, we found that there are 21 and 81 classes in `PascalVOC-SP` and `COCO-SP`, respectively, and the variability of the label class in each graph is very limited. For example, in general, it is not possible to have 21 or 81 different classes via a single graph within `PascalVOC-SP` and `COCO-SP`, respectively. To have sufficient training of the GNNs, the graph adjacency matrices are allocated into one big adjacency matrix in a block-wise manner (Figure. 5). Through this manner, the train, validation, and test split can be conducted.

### C.3.3 TRAINING DETAILS

To address the inherent class imbalance, we use weighted cross-entropy (WCE) loss function, denoted as

$$\text{WCE} = -\sum_{i=1}^{N} w_{y_i} \log(p_{y_i}), \tag{32}$$

in which we let $y_i$ be the true class label for the node $i$, $p_{y_i}$ is the predicted probability for the true class and $w_{y_i}$ is the weight assigned to the true class. WCE adjusts the penalty applied to misclassified examples based on their class frequency. Specifically, the class weights are computed inversely to their frequency in the dataset, ensuring that minority classes, which are underrepresented, receive higher penalties during training. This mechanism forces the model to focus not only on the dominant classes but also on those that are less frequent, leading to a more balanced learning process.

For evaluation, we employ the macro F1-score, which calculates the F1-score for each class individually and averages them to give equal weight to each class, regardless of its frequency. This is particularly relevant for imbalanced datasets, where accuracy alone may obscure poor performance in minority classes. By using the macro F1-score, we ensure that the model's ability to correctly classify underrepresented classes is fairly assessed. The evaluation phase begins by feeding the input data into the model, after which the predicted labels are obtained by selecting the class with the highest predicted logit. These predictions are compared to the ground truth, and the macro F1-score is computed to assess the model's overall classification performance, ensuring a balanced evaluation across all classes. Finally, we assign channel-wise normalization to all input data.

### C.3.4 DMD-GNN IMPLEMENTATION AND HYPERPARAMETERS

Recall that the architecture of our proposed DMD-GNNs follow the so-called MLP_in and MLP_out paradigm, in which the spectral filtering process of our model takes place via a lower dimensional feature space induced from the initial embedding from MLP_in. However, this architecture can be slightly modified for LRGBs. This is because the feature dimension of LRGBs is far below the number of nodes in each graph. In this case, the feature dimension, or the actual nonzero number in the learnable filtering diagonal matrix (i.e., $\text{diag}(\boldsymbol{\theta})$ in equation (14) will be even less than $d$. For handling this case, we set the truncation rate of DMD-GNNs as -1 to provide sufficient number of free parameters to $\text{diag}(\boldsymbol{\theta})$. In addition, we also slightly swap the feature propagation steps in DMD-GNNs by doing spectral filtering first, followed by two MLP layers, that is

$$\mathbf{H}(\ell+1) = \boldsymbol{\Psi}\text{diag}(\boldsymbol{\theta})\boldsymbol{\Psi}^{\top}\mathbf{H}(\ell)\mathbf{W}(\ell), \quad \text{and} \quad \mathbf{H}(\text{out}) = \text{MLP}_2(\text{MLP}_1(\mathbf{H}(\ell+1)), \tag{33}$$

where the functionality of $\text{MLP}_1$ and $\text{MLP}_2$ is to gradually increase the dimension of $\mathbf{H}(\ell+1)$ to the label dimension. This architecture is essentially equivalent to our original model structure, with the only difference as stacking two MLPs after the convolution. This change shows better results compared to the original one, we note that this may be due to that compared to classic node classification tasks, tasks on LRGBs require a more fluent information transaction over graphs, and MLP gives additional feature propagation over the self-loops of the nodes, that is $\mathbf{HW} = \text{diag}(\mathbf{1})\mathbf{HW}$. Furthermore, instead of reducing the feature dimension, two subsequent MLPs provide higher free learnable parameters for the model. In terms of model training, due to the relatively large scale of LRGBs, for all DMD GNNs, we fixed the learning rate as 0.0005, weight decay as $5\text{e}^{-3}$,

dropout = 0.1, and the hidden dimension as 256, we also fixed the batch size as 128 for training set, 500 for both validation and test set. In addition, all models have 10 layers to ensure that the receptive field of the initial node can cover the long-range nodes.

### C.3.5 FURTHER DISCUSSION

Although DMD-GNNs show good results via both `COCO-SP` and `PascalVOC-SP` datasets, the performances are still relatively underscore compared to those methods (Topping et al., 2021; Karhadkar et al., 2022) that are specifically designed for handling LRGBs. In addition, as we mentioned, our proposed DMD-GNNs are not specifically designed to accomplish LRGBs, but rather a generic methods to enhance GNNs in various tasks. Moreover, as we mentioned in the data pre-processing section, due to the large (total) number of nodes and edges, we only selected 10% of the dataset for training our model, and this will further limit our model's performance.

### C.4 SPATIAL-TEMPORAL DYNAMIC PREDICTIONS

#### C.4.1 DATASETS AND DATA PRE-PROCESSING

We select three spatial-temporal graph datasets from `torch-geometric-temporal` (Rozemberczki et al., 2021b). Below we describe their basic information from torch geometric temporal.

**CHICKENPOX** `Chickenpox` is a dataset of county-level chickenpox cases in Hungary between 2004 and 2014. We made it public during the development of PyTorch Geometric Temporal. The underlying graph is static - vertices are counties and edges are neighborhoods. Vertex features are lagged weekly counts of the chickenpox cases (we included 4 lags). The target is the weekly number of cases for the upcoming week (signed integers). Our dataset consists of more than 500 snapshots (weeks).

**COVID** `Covid` (Panagopoulos et al., 2021) is a dataset of mobility and history of reported cases of COVID-19 in England NUTS3 regions, from 3 March to 12 of May. The dataset is segmented in days and the graph is directed and weighted. The graph indicates how many people moved from one region to the other each day, based on Facebook Data For Good disease prevention maps. The node features correspond to the number of COVID-19 cases in the region in the past window days. The task is to predict the number of cases in each node after 1 day. For details see this paper

**WIKIMATH** `Wikimath` is a dataset of vital mathematics articles from Wikipedia. We made it public during the development of PyTorch Geometric Temporal. The underlying graph is static - vertices are Wikipedia pages and edges are links between them. The graph is directed and weighted. Weights represent the number of links found at the source Wikipedia page linking to the target Wikipedia page. The target is the daily user visits to the Wikipedia pages between March 16th 2019 and March 15th 2021 which results in 731 periods.

In terms of the data pre-processing, different from other experiments, we did not conduct feature and adjacency normalization, based on the default setting of the dataloader. In addition, for a more appropriate comparison to show the DMD's advantages of the graph structure estimation, we also involved KNN graphs for some baselines, resulting in GCN-KNN and GAT-KNN. We generate the KNN graph following the setting in (Wu et al., 2023a). Similar to the settings in the LRGBs experiment, we adapt the model architecture as expressed in equation (33), and set $\xi = -1$ same as the one used in the initial node classification experiment. For all experiments in STGs, we let learning rate to be 0.0005 and weight decay be 0.0001 with dropout as 0.8, and hidden dimension as 64. The figure 6 below shows the difference in terms of graph adjacencies from the original graph (`Chickenpox`) and DMD estimates.

#### C.4.2 INFLUENCE ON INFECTIOUS DISEASE EPIDEMIOLOGY

The estimated data-driven graph adjacency matrix from DMD plays an important role in terms of infectious disease modeling. Specifically, the **SIR model**, which divides a population into three compartments: Susceptible ($S(t)$), Infected ($I(t)$), and Recovered ($R(t)$). The basic form of the SIR

Normalized Adjacency Matrix

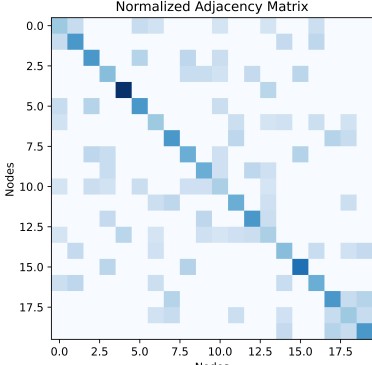

DMD-based Adjacency Matrix

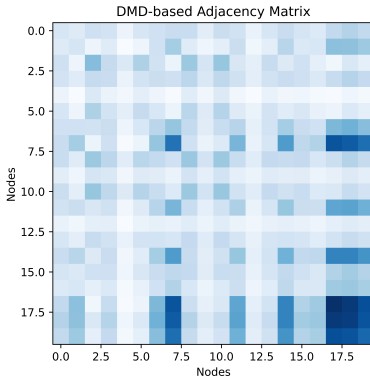

Figure 6: Heat plots of the original graph adjacency matrix (chickenpox) and DMD reconstructed adjacency matrix. One can check that although with the same number of eigenvectors (i.e., $\xi = -1$), compared to the original graph adjacency, DMD is able to deliver a dense adjacency in a data-driven manner, e.g., historical records.

model is as follows:

$$\frac{dS}{dt} = -\beta S(t)I(t) \quad \frac{dI}{dt} = \beta S(t)I(t) - \gamma I(t), \quad \frac{dR}{dt} = \gamma I(t), \tag{34}$$

where:

- $\beta$ is the *transmission rate*, representing the rate at which susceptible individuals become infected through contact with infected individuals;
- $\gamma$ is the *recovery rate*, representing the rate at which infected individuals recover and move into the recovered category.

We note that the SIR model we instanced here requires the dynamic of the disease spread in a closed population, in analogy to the diffusion process, in which we assume the physics shall be operated with limited space. In addition, we will also assume that all individuals have the same probability to contracting with each other with any addition constraints. The transmission dynamics between nodes can be further described using an adjacency matrix $\mathbf{A}$, where $\mathbf{A}_{ij}$ represents the transmission potential between nodes (individuals) $i$ and $j$. Accordingly, a spatial relationship based SIR model can with the form as:

$$\frac{dS_i}{dt} = -\beta S_i \sum_j \mathbf{A}_{ij} I_j, \quad \frac{dI_i}{dt} = \beta S_i \sum_j \mathbf{A}_{ij} I_j - \gamma I_i, \quad \frac{dR_i}{dt} = \gamma I_i. \tag{35}$$

A static adjacency matrix $\mathbf{A}$ purely relies on geographic adjacency or predefined connections is often insufficient to capture the true dynamics of disease spread. As we have illustrated in Section 5.3, DMD allows us to estimate a *data-driven, time-evolving adjacency matrix* from historical epidemiological data, reflecting changes in transmission dynamics.

## C.5 Link Prediction Details

### C.5.1 Task information and Data Pre-Processing

Link prediction is a binary classification task aimed at identifying whether a connection exists between two nodes (Kipf & Welling, 2016). A pair of nodes with an existing edge is considered a "positive link," categorized as class 1. In contrast, node pairs without a connecting edge are labeled as "negative links," corresponding to class 0. Typically, the prediction process involves two phases: (1) an encoder is used to learn node embeddings based on the positive links; (2) a decoder then predicts the presence of a link by calculating the inner product of the embeddings of node pairs. The dataset is split at the edge level for training, validation, and testing. Specifically, 5% of the edges are randomly chosen as validation data, while 10% are used as test data. The remaining edges form the training set. These selected edges represent the positive links, and the negative links are randomly sampled from pairs of unconnected nodes. The number of negative edges is balanced with the positive ones, allowing accuracy to be used as the evaluation metric.

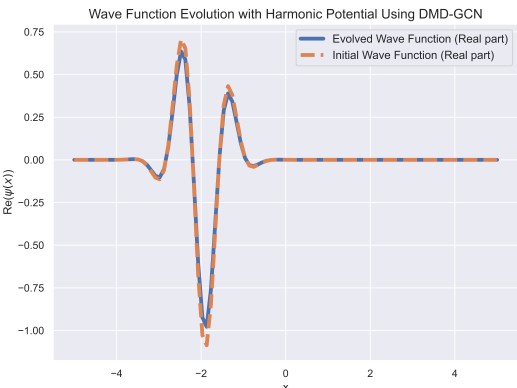

Figure 7: Illustration on the power of PIDMD-GCN to capture the evolution of the quantum wave function. One can see that the estimated (blue) evolution generated from the PIDMD-GCN successfully captured the original evolution of the wave function.

### C.5.2 DMD-GNN IMPLEMENTATION AND HYPERPARAMETERS

In terms of the hyperparameters, since DMD-GNNs serve as the graph encoders in the link prediction task, we fixed most of the hyperparameters the same (including baselines) as the previous node classification tasks, except we let weight decay as 0.005, dropout as 0.7, and number of hidden dimensions as 16. We note that we didn't use the ROC-AUC as the evaluation metric as leveraged in Kipf & Welling (2016), instead, we adopted the average accuracy.

### C.6 DIRECT GRAPH AND PHYSICS INFORMED DMD

We deploy two types of DMD-GNN models via directed and undirected graph `Photo` and `CS`, respectively (Lin & Gao, 2023; Kipf & Welling, 2017). The hyperparameters of both models are exactly the same as the node classification experiment.

### C.6.1 PHYSICS INFORMED DMD AND PIDMD-GNN FOR SCIENCE

The attempt to develop PIDMD can be traced back to the work Baddoo et al. (2023), in which if some characteristics of the operator in the original dynamic are known (e.g., equation (15)), it shall be applied as a constraint to the DMD estimate. These constraints align the model with the fundamental laws (such as symmetry, conservation laws, and self-adjoint properties) governing the system, leading to more reliable and noise-resilient predictions. PIDMD's key advantage lies in its ability to enforce physical consistency, which improves generalization and robustness, especially when applied to well-defined physical systems.

> *PIDMD-GNNs may own the potential as a tool for solving complex dynamic systems.*

We provide a simple example in the following section.

### C.6.2 ILLUSTRATIVE EXAMPLE: SCHRÖDINGER EQUATION

In this example, we briefly discuss the power of PIDMD-GNN in terms of solving the time-dependent Schrödinger equation for a one-dimensional free particle, which can be written as:

$$i\hbar\frac{\partial}{\partial t}\psi(x,t) = -\frac{\hbar^2}{2m}\frac{\partial^2}{\partial x^2}\psi(x,t), \tag{36}$$

where we let $\psi(x,t)$ as the wave function of the particle at the specific position at time $t$, and we further denote $\hbar$ as the Planck constant, and $m$ represents the particle's mass. To make a proper simulation of the equation, we let the number of nodes (particles) as 100, $x_0 = 2$ as the initial position of the particle. The real part of the wave function is set as Gaussian at $x_0$ modulated by the cosine function as the momentum. The imaginary part of the wave function is also a Gaussian by a sine

function, which represents the phase of the quantum system. The real and imaginary parts of the wave function evolve according to the system's symmetric Hamiltonian operator, which can be analyzed as the graph adjacency matrix, ensuring that probability is conserved over time.

In terms of the graph construction, as our example is of a 1D grid, each node is typically connected to its nearest neighborhoods. We leave the construction of the graph adjacency for a high dimensional physical system as a future work. In addition, in this illustration, we did not assign edge weights to the adjacency. Finally, the node features are the real and imaginary parts of the wave function. From Figure. 7, one can clearly see that PIDMD-GCN successfully captured the evolution of the wave function of the system (real part). However, whether PIDMD-GCN remains powerful in a more complex and long-term evolved system is out of the scope of this work, we leave it as future works.

### C.6.3 OPTIMIZATION WITHIN PIDMD

In this section, we briefly review the optimization process involved in both original DMD and PIDMD, our review adapts the work in (Baddoo et al., 2023). We recall that the DMD algorithm is equivalent to finding the solution of the following optimization problem

$$\underset{\text{rank}(\mathbf{K}) \leq d}{\arg\min} \ \|\mathbf{H}(\ell+1) - \mathbf{K}\mathbf{H}(\ell)\|_F. \tag{37}$$

However, when the information of the underlying physics is known, one may adopt the following

$$\underset{\mathbf{K} \in \mathcal{M}}{\arg\min} \ \|\mathbf{H}(\ell+1) - \mathbf{K}\mathbf{H}(\ell)\|_F, \tag{38}$$

where we denote $\mathcal{M}$ as the generic condition for the potential compact manifold that contains the solution of all $\mathbf{K}$. In this case, the problem in equation (38) serves as Procrustes problem, which seeks to find the optimal transformation between two matrices subject to certain constraints on the class of admissible transformations (Schönemann, 1966). The solution to equation (38) can be obtained by adopting the method developed in (Schönemann, 1966) for orthogonality constraint, (Higham, 1988) for symmetric constraint, and (Eldén & Park, 1999) for Stiefel manifold. In addition, one can also control the sparsity of $\mathbf{K}$ by incorporating the regularization of its norm, that is

$$\underset{\mathbf{K} \in \mathcal{M}}{\arg\min} \ \|\mathbf{H}(\ell+1) - \mathbf{K}\mathbf{H}(\ell)\|_F + \frac{1}{2}\|\mathbf{K}\|_2, \tag{39}$$

and this aligns with the recent idea of controlling the sparsity of the transportation plan via optimal transport problem (Tianlin Liu, 2023). Lastly, as we have mentioned in Section 8, we are also interested in assigning the so-called coupling condition to $\mathbf{K}$, yielding

$$\underset{\mathbf{K}\mathbf{1}=\mu, \mathbf{K}^\top \mathbf{1}=\nu}{\arg\min} \ \|\mathbf{H}(\ell+1) - \mathbf{K}\mathbf{H}(\ell)\|_F, \tag{40}$$

and since $\mathbf{H}(\ell+1)$ and $\mathbf{H}(\ell)$ are graph observables, this indicates that the classic OT problem might be able to be interpreted as a GNN convolution via a dense graph (i.e., $\mathbf{K}$). This assumption is reasonable due to the fact that OT finds the optimal solution between distributions, the transportation from the source to the target can be sparse as one-to-one or dense as one-to-many. In more general, under the OT scope, one can even drop the square condition of $\mathbf{K}$, since the number of elements of the source can be different compared to the elements in targets. This could own the potential of defining a new branch of DMDs with system states via different dimensions. Lastly, if $\mathbf{H}(\ell+1)$ and $\mathbf{H}(\ell)$ are sourced from the different graph, then the above equation means that there shall be a convolution (using $\mathbf{K}$), under the scope of OT, to transport one feature matrix to another. This might be of interest in terms of knowledge graph alignment (Zeng et al., 2021). We leave these exciting topics to the future works.

## D    FINAL REMARKS AND FUTURE RESEARCH DIRECTIONS

**Insufficient Feature Dimension**    In some cases of our initial tests, we found that DMD showed limited performance when the feature dimension of the graph was low. That is, the feature information is not rich enough, and this leads to a limitation of our model and a good standing point for developing more complete DMD variants for dealing with this situation. Fortunately, in most of our included experiments, DMD-GNN performed fairly well compared to the baselines, even with lesser filtering

parameters. This could be due to its data-driven nature to refine the eigenspace (i.e., $\mathbf{\Psi}$) from the system snapshots.

From Figure 4, although the ranks leveraged in different graphs are different, from our empirical observations, graphs with similar characteristics are more likely to share similar truncation rates. For example, for homophily graphs, we observed that $\xi = 0.85$ often achieves the best learning accuracy, whereas for heterophilic graphs, $\xi = 0.7$ is the selected rate for all graphs. This unified quantity of $\xi$ in different types of graphs may be due to the nature of their homogeneity (citation graphs). However, to sufficiently determine the suitable truncation rate for any given dataset is a challenging problem not only for DMD-GNN but also for other SVD-related tasks. We kindly leave it to the future works.

**Information Integrity**    Similar to the above discussion, one can also observe that in tasks where DMD-GNNs serve as an intermediate module for the downstream task, e.g., link prediction, the DMD-GNN-based models sometimes underperform their original model, e.g., GCN vs. DMD-GCN. This phenomenon leads us to explore the truncation process in DMD in the future, or in other words, what information shall be preserved/ignored. We suppose this may heavily depend on the type of downstream tasks, e.g., whether the encoded feature generated from DMD-GNNs is still distinguishable from each other via the classification tasks. Furthermore, the construction of the node feature differs between datasets, making it even more difficult to sufficiently quantify and analyze the above challenge. We suppose the analysis could merely conducted on the synthetic datasets (e.g., from the Contextual Stochastic Block Model (CSBM)) where features are sampled from pre-defined distributions, and the results shall be statistical-related (Wu et al., 2023b).

**Graph Classification and Expressive Power**    Although our experiment did not run into the graph classification tasks, one can still check that our model outperformed the GIN model with is proved to have maximal expressive power (Xu et al., 2019b) via the node classification task (e.g., Table 1.) In addition, another reason for the absence of the graph classification is due to the fact that similar to the settings in link prediction, in graph classification, one can also consider DMD-GNNs as an encoder prior to those so-called readout layers such as maximal and average pooling (Zhang et al., 2023). Similar to our discussion on the information integrity paragraph, we shall be focusing on the interactions between DMD-GNN outputs and the downstream calculations. Although DMD-GNNs can inherit many advantages from the original GNN dynamics, fully quantifying its impact on various downstream tasks requires an ideal control of the settings of each module of the model and various data inputs. We aim to provide new insights into this direction. Such investigations will likely lead to novel designs that further enhance DMD-GNN's capabilities for a broader range of graph-based learning tasks.

**Adopting DMD to More Complex Dynamics**    Another interesting field in which DMD-GNNs can be applied is the learning tasks on physical and chemical dynamics, e.g., molecular dynamics (Xu et al., 2024) and quantum (or standard) many-body problems (Satorras et al., 2021), in which the main task is to simulate and predict the particles' position and feature changes through the evaluation of the system. Many recently developed methods tend to construct the graph (e.g., for the molecules) via a combination of KNN and its original topology, and such graph connectivity remains unchanged through the rest of the simulation/prediction. Although many physical/chemistry inductive biases have been introduced, a data-driven approach that generates a suitable graph structure that fits the propagation defined in the relevant models remains insufficiently explored. A deeper reason for this could refer to the changes in the compound structure due to the changes in its energy (e.g., Gibbs free energy). This alignment may lead to a deeper insight into the relationship between numerical data-driven methods in machine learning and actual chemical/physical laws. We also expect to conduct our research in this field in our future research.

