# OpenReview forum: "When Graph Neural Networks Meet Dynamic Mode Decomposition"
_ICLR.cc/2025/Conference — ICLR 2025 Poster_

### Official Review · Reviewer_EfEy · 2024-10-31

**Soundness:** 3
**Presentation:** 3
**Contribution:** 3
**Rating:** 8
**Confidence:** 3

**Summary:**

The authors investigate GNNs from the perspective of Koopman theory. They use dynamic mode decomposition (DMD) to approximate the Koopman operator with a finite dimensional matrix. Based on this they propose Graph Neural Network (GNN) models using the low-rank eigenfunctions given by DMD. They compare their model to state of the art GNNs in extensive experiments on several learning tasks, in which the DMD GNNs achieve competitive results.

**Strengths:**

1) The background of Koopman theory and DMD is well explained in  section 3 and 4.
2) The authors made extensive experiments and effort to validate their models.
3) To the best of my knowledge the idea presented is novel.

**Weaknesses:**

Major:

1) While sections 3 and 4 are good to understand, I am not sure, if I understood sections 5 and 6 about the model itself completely. A pseudocode and a more detailed description of Figure 1 could help.
2) The proposed model is not compared to GIN, although it is maximally expressive.
3) In line 425,426 you state that DMD-GNNs “outperform the baseline models across the majority of datasets”, which is formulated too strong, because in most cases the performance increases are only minor and/or not significant.

Ambiguities:

4) The sentence on attention-based GNNs (l. 90-93) is a little bit confusing
5) in line 411, you are listing ACMP as a baseline model but it is not occurring in the tables.
6) APPNP is not cited anywhere and explained what the abbreviation means

Minor:

7) The related work section is very general, the differentiation to previous work is unclear.
8) The equation in line 133 is probably wrong, because it would lead to K = 1. Should probably be the linearity of K_t?
9) Line 237: K = M*F would mean that M = 1? Probably a mistake?
10) Please briefly mention the results of App. C.2.4 in 7.2
11) KNN abbreviation in line 466 was not defined
12) Line 234: The pseudo-inverse of H(l) has already been defined in line 227, and it is not even occurring in Eq. (11)
13) Please divide Figure 2 in a and b
14) Fig. 2 is much too small
15) First sentence in 7.1 is confusing.

Spelling/Grammar:

16) line 146: underlying
17) Headline 4: How GNNs Resonate with Dynamic Systems
18) Table 1: Caption should state that the results are from node classification experiments. Point is missing at the end of the caption.
19) line  523-524, grammatic error

**Questions:**

a) How many hidden nodes were used in the baseline models for node classification? Can you include a comparison of the number of model parameters?

b) The link prediction VGAE framework is not state-of-the-art, why did you use it and not SEAL?

c). For link prediction you used average accuracy instead of ROC-AUC. Why?

d). Why was ChebNet only used for spatial-temporal dynamics experiments?

e) Why did you use the parameter \xi = 0.85 for homophilic graphs but the sensitivity analysis was perfomed only up to \xi = 0.8?

f). Could it be interesting to use other discretization methods than Euler?

---

> ### Author Response · Authors · 2024-11-22
> **Summary of Response to Reviewer EfEy**
>
> We sincerely thank you for your detailed and thoughtful review, which has been invaluable in improving the quality and presentation of our work.
>
> In response to your comments, we have included pseudocode in Appendix C.1 and revised Figure 1 to provide clearer explanations of our model's design and the role of the DMD module. We also addressed ambiguities, corrected inaccuracies, and refined our claims to ensure they were both accurate and well-supported. Furthermore, we expanded the related work section and explored potential future applications, such as biological and physics-informed systems, as discussed in Appendix D.
>
> Your constructive feedback and supportive comments have encouraged us to further refine our paper, and we deeply appreciate the time and effort you have dedicated to reviewing our work. Thank you once again!

---

> > ### Author Response · Authors · 2024-11-22
> > **Detailed Response Part 1**
> >
> > **Comment: While sections 3 and 4 are good to understand, I am not sure, if I understood sections 5 and 6 about the model itself completely. A pseudocode and a more detailed description of Figure 1 could help.**
> >
> >
> > **Reply:** Thank you very much for your suggestion. Based on your suggestion, in our revised submission, we have included the pseudocode for our model via Appendix C.1.
> >
> > **Comment: The proposed model is not compared to GIN, although it is maximally expressive**
> >
> > **Reply:** Thank you very much for your insightful suggestion. Based on the best of our knowledge, GIN is designed for the purpose of graph-level classification tasks due to its maximal expressive power to the WL tests. Although we can also deploy the DMD algorithm to the snapshots provided by GIN, resulting in the DMD-GIN model, the functionality of DMD-GIN will be challenging to explore compared to those tasks (e.g., node classification) using DMD-GNN output directly for prediction, since one need to at least apply some pooling strategies to the DMD-GNN outputs to accomplish the graph classification. In this case, DMD-GNN is similar to its functionality via link prediction, where its output only serves as one intermediate step of the whole model.
> >
> > Nevertheless, based on your suggestion, we include some discussion in **Appendix D**, via the paragraph: **Graph Classification and Expressive Power**, in which we highlight challenging aspects and potential direction for the future study that is undergoing by us.
> >
> > **Comment: In line 425,426 you state that DMD-GNNs “outperform the baseline models across the majority of datasets”, which is formulated too strong, because in most cases the performance increases are only minor and/or not significant.**
> >
> > **Reply:** Thank you very much for pointing this out. In our revised submission, we have moderated our description below:
> >
> > *One can check that DMD-GNNs consistently achieve notable improvements over baseline models across the majority of datasets.*
> >
> > We also checked the description to ensure our claim is accurate. We hope our changes can meet your expectations on these comments.
> >
> > **Comment: The sentence on attention-based GNNs (l. 90-93) is a little bit confusing**
> >
> > **Reply:** Thank you very much for spotting this out. In our revised submission, we have clarified this content below:
> >
> > *In addition, attention-based GNNs (e.g., GAT) and transformer-based diffusion models (e.g., Difformer) compute node-level attention scores based on feature similarities within each layer (i.e., the current system state) but rarely consider cross-layer feature interactions, which could better capture relationships between multiple system states*
> >
> > **Comment: In line 411, you are listing ACMP as a baseline model but it is not occurring in the tables.**
> >
> > **Reply:**  We sincerely apologize for this careless mistake! We have added ACMP back to our revised submission.**
> >
> >
> > **Comment: APPNP is not cited anywhere and explains what the abbreviation means.**
> >
> > **Reply:** Thank you for posting this out. In our revised submission, we have cited APPNP on our main page. Also, the propagation of APPNP was originally included in Appendix B via equation (22).
> >
> > **Comment (Minor): The related work section is very general, the differentiation to previous work is unclear.**
> >
> > **Reply:** Thank you very much for your suggestion. Based on your comments, we have tried our best to revise the related work section, especially on the related work of the graph neural diffusion models. Please take a look at our revised submission for further details. In addition, we wish to highlight that although we have maximized our effort to link the DMD algorithm to the current machine learning approaches via some recent works, the area of incorporating DMD to GNNs seems still in its infancy, causing limited comparison between the current machine learning works via the scope of DMD in the second paragraph of related work. In the future study, we also wish to deploy this powerful tool to other machine-learning areas.
> >
> > **Comment (Minor): The equation in line 133 is probably wrong because it would lead to K = 1. Should it probably be the linearity of $K_t$?**
> >
> > **Reply:** Thank you very much again for spotting this. You are definitely right. To illustrate the linearity of the operator $\mathcal K_t$, the expression shall be $$\mathcal{K}_t(a\boldsymbol{\phi}_1 +b \boldsymbol{\phi}_2) = a \mathcal K_t\boldsymbol{\phi}_1+b \mathcal K_t\boldsymbol{\phi}_2$$
> >
> > We have included the changes in our revised submission.
> >
> > **Comment (Minor):  Line 237: K = M*F would mean that M = 1? Probably a mistake**
> >
> > **Reply:** That is definitely a mistake. We are sorry for this. The expression of $\mathbf K$ shall refer to equation (12). In case of any confusion in the future, we have deleted that equation in our revised submission. Thanks for this careful check.

---

> > > ### Author Response · Authors · 2024-11-22
> > > **Detailed Response Part 2**
> > >
> > > **Comment (Minor):  Please briefly mention the results of the App. C.2.4 in 7.2**
> > >
> > > **Reply:** Thank you very much for your suggestion. Our original intention is to refer the reader to read Appendix C.2.4 to see the slight change in the model's architecture for fitting the long-range graph datasets. We apologize for this misunderstanding. In our revised submission, we have changed the content to
> > >
> > > *In Appendix C.2.4, we show how we slightly changed our model's architecture to better handle this phenomenon.*
> > >
> > > We hope our clarification can meet your requirements to resolve this concern.
> > >
> > > **Comment (Minor): KNN abbreviation in line 466 was not defined**
> > >
> > > **Reply:** Thank you again; we have added the relevant abbreviation for KNN in our revised submission.
> > >
> > > **Comment (Minor): Line 234: The pseudo-inverse of H(l) has already been defined in line 227, and it is not even occurring in Eq. (11)**
> > >
> > > **Reply:** Sorry about this. In our revised submission, we have deleted this repetitive part.
> > >
> > > **Comment (Minor): Please divide Figure 2 in a and b and Fig. 2 is much too small.**
> > >
> > > **Reply:** Thank you very much for your suggestion. In our revised submission, we have tried our best to increase the size of Figure 2. Also, we have added index (a) and (b) to the figure. Due to the space and time limit, we are still adjusting the size of the figure to make it better; we promise to deliver a satisfactory outcome via the camera-ready process.
> > >
> > > **Comment (Minor): The First sentence in 7.1 is confusing.**
> > >
> > > **Reply:** Thank you for this detailed suggestion. In our revised submission, we have streamlined the first sentence in Section 7.1 (as the concept of the homophilic and heterophilic has been introduced in the previous sections) to prevent any further confusion. We hope the revised version is clearer and addresses your concerns.
> > >
> > > **Question: How many hidden nodes were used in the baseline models for node classification? Can you include a comparison of the number of model parameters?**
> > >
> > > **Reply:** Thank you very much for your suggestion. Here, we show one parameter comparison case between GCN (two layers), ChebNet (one layer), and DMD-GCN (one layer) in Cora. We let the hidden dimension be 16. Accordingly, the total parameters for GCN are $1433\times 16 + 16\times 7 + 16 + 7 = 23063$ (with bias), and for ChebNet with polynomial order of 2 is $2\times 1433 \times 16 + 2 \times 16 \times 7 + 17 + 7 = 46103$. In terms of DMD-GNNs, as we adopt a spectral filtering paradigm, the number of parameters is $1433 \times 16 + 421 + 16 = 23365$, where $421$ is the number of actual ranks used in DMD-GCN. One can check the number of parameters of DMD-GCN, which is similar to GCN but less than ChebNet. In response to your suggestion, we have included relevant content in Appendix C.2.1. Thank you very much for making our paper more complete.
> > >
> > > **Question: The link prediction VGAE framework is not state-of-the-art; why did you use it and not SEAL?**
> > >
> > > **Reply:** Thank you very much for your question. We highlight that our motivation for including the link prediction (LP) experiment is to illustrate that DMD-GNN can be used as a graph encoder for the downstream link prediction task. Meanwhile, we also want to show that DMD-GNN can achieve almost identical or even higher prediction performances than its original GNN counterpart in LP. Fortunately, DMD-GNN did a good job of supporting our claim. Accordingly, we did not select the current SOTA framework.
> > >
> > > **Question: For link prediction, you used average accuracy instead of ROC-AUC. Why?**
> > >
> > > **Reply:** Thank you very much for your question. We use accuracy as the metric instead of AUC-ROC because we adopt equal-proportion sampling for positive and negative edges. We have included this clarification in our revised submission.
> > >
> > > **Question: Why was ChebNet only used for spatial-temporal dynamics experiments?**
> > >
> > > **Reply:** Thank you for the comments. In our revised submission, we have included ChebNet in the main node classification tasks to verify that the refined spectral domain enhances DMD-GNN, leading to improved learning outcomes. Additionally, we have enriched the discussion of the results, making the comparison between DMD-GNN and GNN more comprehensive and thorough, providing a deeper understanding of the advantages of our approach.

---

> > > > ### Author Response · Authors · 2024-11-22
> > > > **Detailed Response Part 3**
> > > >
> > > > **Question: Why did you use the parameter $\xi = 0.85$ for homophilic graphs, but the sensitivity analysis was performed only up to $\xi = 0.8?$**
> > > >
> > > > **Reply:** We thank the reviewer for the comment. The sensitivity analysis on $\xi$ is motivated to show the trend on how DMD-GNNs adaptation power on different types of graphs changed by the quantity of $\xi$, not for showing what is the value of the optimized $\xi$. As we can see from Figure 2 (left), in Cora, the performance of DMD-GNN models increases via the increase of $\xi$ from 0.2 to 0.8 since a higher $\xi$ indicates that the model utilizes a higher amount of spectral information, which is typically useful for homophilic graphs. Accordingly, we suppose the current figure illustrates our initial purpose. However, if you think including $\xi = 0.85$ or another number is necessary, we are happy to further include it at any stage. Thanks again.
> > > >
> > > > **Question: Could it be interesting to use other discretization methods than Euler?**
> > > >
> > > > **Reply:** Yes, we can. We can follow the implementation of GRAND [1] to use advanced discretization methods, such as leapfrog and Runge–Kutta methods etc.
> > > >
> > > > [1]: GRAND: Graph Neural Diffusion.
> > > >
> > > >
> > > > **Lastly, following your instructions, we have carefully checked the Spelling/Grammar errors in our paper, followed by a few rounds of extra proof-readings; thanks again for helping us to make our paper better!**

---

> > ### Comment · Reviewer_EfEy · 2024-11-22
> > **Response to new version**
> >
> > Dear authors,
> >
> > thank you for thoroughly revising your paper, I believe your changes have been very beneficial.
> > You have addressed all my minor concerns and most of my major concerns, especially the pseudocode helped a lot.
> >
> > Regarding major point 2, I think you have misunderstood my comments. I did not mean to ask why you did not use GIN in your model, but why you did not compare against it, because it is proven to be maximally expressive and might solve the learning tasks in your experiments better. You are right, that GIN is often used in graph classification, but it would still be an interesting baseline. However I will not insist on the comparison, since the strength of you manuscript is a new theoretical framework, not performance of an algorithm.
> >
> > Regarding major point 3, In my mind 'notable improvements over baseline models' says the exact same thing as before.

---

> ### Author Response · Authors · 2024-11-22
> **Reconsideration of the Score**
>
> We deeply value the time and effort you have invested in reviewing our work and welcome any further suggestions to make the paper even better. If you find that our responses and revisions meet your expectations, we kindly ask you to consider increasing your score to reflect the improvements made. Your support would greatly encourage us to continue advancing this research direction. Thank you once again!

---

> ### Author Response · Authors · 2024-11-22
> **Repose to Reviewer's Comment**
>
> Dear Reviewer:
>
> Thank you very much for your prompt and valuable feedback. Following your suggestions, we have made the following revisions:
>
> * GIN has been added to the experiment for the node classification experiment in Table 1.
>
> * Formulation and brief introduction of GIN has been added to **Appendix B** (equation 31).
>
> * We keep the paragraph **Graph Classification and Expressive Power** in **Appendix D** and briefly mention the comparison of node classification results between our model and GIN.
>
> Additionally, to better reflect the comparison between our model and the baselines, we have revised the description of the node classification results to:
>
> *One can check that DMD-GNNs demonstrate comparable performance to baseline models, with modest improvements on some datasets.*
>
> We suppose these changes address your concerns and improve the clarity and completeness of our manuscript. Please let us know if you need any further revisions.
>
> Thanks again!
>
> Authors

---

> ### Author Response · Authors · 2024-11-26
> **Follow up Response**
>
> Dear Reviewer:
>
> Thank you for taking the time and effort to comment on and discuss our work with us. We truly believe that the quality of our work has been significantly improved based on your suggestions.
>
> Given the approaching deadline for submitting the revised manuscript, could you please have a look at our responses on the inclusion of the GIN model and the rephrasing of our experimental results? In addition, if there are any further changes that you want us to add, we are standing by now :)  If you feel our response has addressed your concerns to a satisfactory extent, we would be grateful if you could consider revising your score. Your support and acknowledgment would mean a great deal to us and encourage us to further explore this fascinating field. Thank you again.
>
> Sincerely, The authors.

---

> > ### Comment · Reviewer_EfEy · 2024-12-02
> > **Reply to revision**
> >
> > Dear authors,
> > thank you very much for the additional changes, I have increased my final rating accordingly.

---

> > > ### Author Response · Authors · 2024-12-02
> > > **Thank you very much**
> > >
> > > Dear Reviewer:
> > >
> > > Thank you very much for your support! We also wish you all the best in your future research!
> > >
> > > Authors

---

### Official Review · Reviewer_WY5R · 2024-10-31

**Soundness:** 3
**Presentation:** 3
**Contribution:** 3
**Rating:** 6
**Confidence:** 2

**Summary:**

This paper provided a novel connection between dynamical mode decomposition and graph neural network. Starting from considering the feature propagation as a dynamical system, authors took advantages of Koopman theorem and DMD in practice to refine this process. They also provided theoretical analysis to state the properties of the model. The results showed consistent improvement of combining the existing models and the DMD module.

**Strengths:**

This paper provided enough details and derivation to explain the proposed model.
The results proved that with refined features, GNN performance can be improved. Therefore, physics-informed or biological-plausible systems can be further developed based on this structure.

**Weaknesses:**

1. Figure 1 is actually confusing as an illustration of the model. The same arrow feels like an input to the next function, and it doesn't really show where DMD module is applied.
2. The motivation of this paper should be expanded. As authors stated at the beginning "a carefully analyzed and refined dynamic can potentially enhance GNN performance by providing deeper insights into feature propagation over graphs", it would be great to see how the features are refined and why it benefits the performance.
3. It would be much clearer if authors could highlight the algorithm of the model.

**Questions:**

1. In Koopman theorem, it states that "a linear process can be found in the infinite-dimensional space", while for the proposed model, the authors used DMD method to imitate this, so only a finite dimensional space is constructed. It means that some information will be ignored. Will this method affect the performance of a more complex dataset? In your experiments, are the ranks different in different datasets? How do you decide the number of filters?

2. In the result table 3, when you compare the results of GCN and DMD-GCN, one can observe that the performance is not improved in all columns. Can you have some additional explanations on what features benefit or "destroy" the model? As I pointed out in the weakness, the model's motivation should be explained by the results.

---

> ### Author Response · Authors · 2024-11-22
> **Summary of Response to Reviewer WY5R**
>
> We sincerely thank you for your detailed feedback and for recognizing the strengths of our work. Your constructive comments have been instrumental in helping us improve both the clarity and the depth of our paper.
>
> In response to your valuable suggestions, we have revised the figures, particularly Figure 1, to better illustrate where the DMD module is applied. We also expanded the motivation in our Introduction section and clarified how DMD refines GNN dynamics to enhance performance. To further support reproducibility and clarity, we included pseudocode for both DMD and DMD-GNN in Appendix C.1. Additionally, we addressed questions about the finite-dimensional nature of DMD, the impact of truncation rates across datasets, and performance variations observed in specific cases.
>
> We deeply appreciate your acknowledgment of the potential for applying our structure to physics-informed and biologically plausible systems, which aligns closely with our future research plans. Thank you for your insightful suggestions, which have helped us further refine the paper and highlight its broader applicability. We hope our revisions meet your expectations and provide a clear picture of our contributions.

---

> > ### Author Response · Authors · 2024-11-22
> > **Detailed Response Part 1**
> >
> > **Comment (from Strength): physics-informed or biological-plausible systems can be further developed based on this structure.**
> >
> > **Reply:** We thank the reviewer for this positive feedback. In fact, this was one of our initial plans to incorporate DMD into the biological or physical fields, such as molecular dynamics, ligand binding, quantum systems, etc. We did not delve into it as the models designed for these areas usually come with some extra conditions, such as equivariant [1], and one may require some additional inductive bias in physics and chemistry via the model developing process. Most importantly, existing models incorporate the message-passing paradigm using both positional and feature information, and this information interacts with each other via propagation. Please refer to the model formulation in [1] for more details. If this is the case, it might be possible to define a generalized DMD algorithm that acts on two interacted domains, i.e., positional and feature. Such DMD has the potential to analyze the complex EGNN [1] and its variants, and the corresponding new models (if available) could show better prediction accuracy via different aspects. We will dive into these exciting fields in our future research.
> >
> >
> > Finally, we have added the relevant content in one newly included section, Appendix D, for further discussion. Please kindly visit our revised submission for more details, and let us know if you need more information. Thanks again.
> >
> > [1]: E(n) Equivariant Graph Neural Networks.
> >
> > **Comment: Figure 1 is actually confusing as an illustration of the model. The same arrow feels like an input to the next function, and it doesn't really show where DMD module is applied.**
> >
> > **Reply:** Thank you very much for pointing this out. In our revised submission, we have revised Figure 1 to explicitly show that DMD takes two system states (snapshots) to produce DMD modes. Thank you very much for helping us to make our paper clearer.
> >
> > **Comment: The motivation of this paper should be expanded. As the authors stated at the beginning, "a carefully analyzed and refined dynamic can potentially enhance GNN performance by providing deeper insights into feature propagation over graphs." it would be great to see how the features are refined and why they benefit the performance.**
> >
> > **Reply:** Thank you very much for this great suggestion. As a following-up, we have made the following changes in our revised submission.
> >
> > *From the dynamical systems perspective, a carefully analyzed and refined dynamic can potentially enhance GNN performance by providing deeper insights into feature propagation over graphs. However, to tackle more challenging tasks, recent GNNs often adopt advanced physical dynamics [1], increasing complexity and hindering interpretability. Thus, a well-established tool is needed to analyze these dynamics and provide deeper insights.*
> >
> > In addition to these changes, we also tried our best to make the rest of this paragraph more explicit in illustrating the impact of applying the DMD algorithm to analyze the GNN dynamic. For your convenience, we also include them below:
> >
> > *Specifically, DMD approximates the Koopman operator with finite-dimensional representations only via the states (i.e., snapshots) of the system, regardless of its complexity. In addition, DMD produces the so-called DMD-modes, which are a collection of refined eigenbasis of the system's original operator (e.g., graph adjacency matrix) with potentially lower-dimensional. These modes capture the principal components driving the underlying GNN dynamics and offer a data-driven adjusted domain for spectral filtering. Building on these advantageous properties of DMD, in this work, we propose DMD-enhanced GNNs (DMD-GNNs). We show that our proposed DMD-GNNs not only can capture the characteristics of the original dynamics in a linear manner but also have the potential to adopt more complex tasks (e.g., long-range graphs [2]) in which the original model of DMD-GNNs usually delivers weak performance.*
> >
> > [1] From continuous dynamics to graph neural networks: Neural diffusion and beyond.
> >
> > [2] Long range graph benchmark.

---

> > > ### Author Response · Authors · 2024-11-22
> > > **Detailed Response Part 2**
> > >
> > > **Comment: It would be much clearer if authors could highlight the algorithm of the model.**
> > >
> > > **Reply:** Thank you very much for your suggestion. We totally agree that we need one additional pseudocode of our proposed methods to explicitly show its steps. Accordingly, we have included the pseudocode for both DMD and DMD-GNNs in Appendix C.1 via our revised submission. Due to the space limit and the website formatting restriction, please kindly check our revised submission. Thanks again.
> > >
> > >
> > > **Question: In Koopman theorem, it states that "a linear process can be found in the infinite-dimensional space", while for the proposed model, the authors used DMD method to imitate this, so only a finite dimensional space is constructed. It means that some information will be ignored. Will this method affect the performance of a more complex dataset? In your experiments, are the ranks different in different datasets? How do you decide the number of filters?**
> > >
> > > **Reply:** Thank you very much for this valuable question. In some extreme cases of our initial tests, we found that DMD showed limited performance when the feature dimension of the graph was low. That is, the feature information is not rich enough, and this leads to a limitation of our model and a good standing point for developing more complete DMD variants for dealing with this situation. We appreciate your insightful observation! Fortunately, in most of our included experiments, DMD-GNN performed fairly well compared to the baselines, even with lesser filtering parameters. This could be due to its data-driven nature to refine the eigenspace (i.e., $\boldsymbol{\Psi}$) from the system snapshots.
> > >
> > > In addition to the actual ranks used for each dataset, we highlight that in Appendix C.2 of our original submission, we have visualized the truncation process via Figure 3 (Now Figure 4). From Figure 3, although the ranks in different graphs are different, from our empirical observations, graphs with similar characteristics are more likely to share similar truncation rates. For example, for homophily graphs, we observed that $\xi = 0.85$ often achieves the best learning accuracy, whereas for heterophilic graphs, $\xi = 0.7$ is the selected rate for all graphs. This unified quantity of $\xi$ in different types of graphs may be due to the nature of their homogeneity. In addition, the truncation rate for the homophilic graph is larger than the one in the heterophilic graph due to the fact that in homophilic graphs, the node connectivity information is of higher importance than in heterophilic graphs as connected nodes are often with similar labels. Accordingly, one shall prefer to include more spectral information for homophilic graphs by leveraging a higher $\xi$ than heterophilic graphs.
> > > However, to sufficiently determine the suitable truncation rate for any given dataset is a challenging problem not only for DMD-GNN but also for other SVD-related tasks. We kindly leave it to the future works.
> > >
> > > Lastly, we have included the discussion above in one new Section (Appendix D) in our revised submission; we sincerely thank the reviewer again for your effort in making our paper better!

---

> > > > ### Author Response · Authors · 2024-11-22
> > > > **Detailed Response Part 3**
> > > >
> > > > **Question: In the result table 3, when you compare the results of GCN and DMD-GCN, one can observe that the performance is not improved in all columns. Can you have some additional explanations on what features benefit or "destroy" the model? As I pointed out in the weakness, the model's motivation should be explained by the results.**
> > > >
> > > > **Reply:** Thank you very much for your insightful question. We suppose this question can related to what information shall be preserved/ignored. Although DMD-GNN consistently preserves or even surpasses the performance of its original dynamic, these observations have usually been found in the case when the output of the tasks is purely generated from DMD-GNNs. However, for tasks like link prediction, where DMD-GNNs are used as an intermediate module, determining whether the DMD output should retain the complete feature information itself becomes a challenging problem. We suppose this may heavily depend on the type of downstream tasks, e.g., whether the encoded feature generated from DMD-GNNs is still distinguishable between each other via the classification tasks. Furthermore, as the construction of the node feature differs between datasets, making it even more difficult to sufficiently quantify and analyze the above challenge, we suppose the analysis could merely conducted on the synthetic datasets (e.g., From Contextual Stochastic Block Model (CSBM)) where features are sampled from pre-defined distributions and the results could be statistical-related, such as this non-asymptotic analysis for over-smoothing [3].
> > > >
> > > > Nevertheless, in Appendix D.2, we include additional discussions on this matter via the paragraph: **Information Integrity**. Again, we sincerely thank you for making our paper better.
> > > >
> > > > [3]: A Non-Asymptotic Analysis of Oversmoothing in Graph Neural Networks.

---

> > > > > ### Author Response · Authors · 2024-11-22
> > > > > **Reconsider the score**
> > > > >
> > > > > We finally take this chance to thank the reviewer again. Your thoughtful feedback has already played a crucial role in refining our work, and we deeply value your expertise and input. Should you have any remaining questions or suggestions, we would be delighted to engage further to ensure the paper meets the highest standards. Meanwhile, we sincerely expect that you can increase your score, given the efforts we have made to address your concerns and the revisions implemented in the paper. Thanks in advance.

---

> ### Author Response · Authors · 2024-11-26
> **Follow up Response**
>
> Dear Reviewer,
>
> Thank you very much for your time and effort in reviewing our work. Your constructive suggestions and insightful questions have been invaluable in significantly improving the quality of our paper.
>
> As the deadline for submitting major revisions is approaching, we have carefully addressed all your concerns to the best of our ability. Could you kindly review our responses and let us know if there are any further changes or improvements we can make?
>
> We truly appreciate your continued support and guidance throughout this process. If you feel our responses have adequately resolved your concerns, we would be grateful if you could reconsider our score.
>
> Sincerely, The authors.

---

### Official Review · Reviewer_33PK · 2024-11-03

**Soundness:** 4
**Presentation:** 4
**Contribution:** 3
**Rating:** 8
**Confidence:** 2

**Summary:**

This work draws a connection between the feature dynamics in GNNs to DMD (a numerical method for Koopman theory). They use this connection to build a novel GNN architecture, which can be used across tasks --- node classification, node regression, link prediction --- and data settings. They validate this architecture with extensive numerical experiments.

**Strengths:**

The paper is generally well written. I particularly like the high level layout; the section titles lead the reader along nicely; 'How GNNs Resonates with Dynamic Systems' is a particularly nice one :)

I like that it brings the solidity of DMD and merges it with a learning architecture; which I believe to be mostly original (although out of my expertise). I think it could be useful in real physical systems. I understand the authors use publicly available and widely known datasets out of necessity in proceedings like this, but I would think the work would present even stronger if there were physical systems datasets to validate on. See more on this in 'Questions'.

**Weaknesses:**

**Clarity of Derivation for Non-Experts**
I did have a bit of trouble following the derivations in Section 4 and 5; I am not a super GNN expert, nor a dynamics systems expert, so this could be on me. I believe this work will be at a disadvantage in an ICLR-like review process bc it truly lies at the intersection of two fields which are typically distinct. I attempt not to punish this work for this valiant effort/approach, but it would be wise of the authors to attempt to preempt this confusion with generous use of intuitive figures. Perhaps an additional figure outlining arguments section 4. Figure 1 covers some, but not all, of this.

&nbsp;

**Scalablity**
Additionally, the authors claim scalability by showing an experiment on 'OGB-arXiv', but I would like a cleaner argument for scalability. It seems a power (s=2) of the adjacency is used in some experiments. This can present issues for memory and runtime.

&nbsp;

**Runtime**
I would also like to see some actual runtime plots/figures/numbers. Don't feel the need to re-run everything to record the times. Just enough to give the reader a sense of scale; what will it cost me (time, memory, etc) to actually run this?

**Questions:**

**The unique benefit of DMD-GNN**

I am really looking for setting which can show the *unique* benefit of DMD-GNN. The authors have gone through extensive derivations and work to derive DMD-GNN. What can it do that a generic, off-the-shelf, large GNN simply cannot do? What tasks does it make significantly easier (perhaps beyond just the final metric going up a bit). Can the authors think of any applications, and ideally datasets, for this purpose? Perhaps data generated from systems closer to how DMD/Koopman theory is typically used would be a nice direction.

---

> ### Author Response · Authors · 2024-11-22
> **Summary of the Response to Reviewer 33PK**
>
> We sincerely thank you for your positive evaluation and for highlighting the strengths of our work. Your thoughtful feedback and constructive suggestions have been incredibly helpful in further improving the clarity and presentation of our paper.
>
> In response to your comments, we have refined our figures, enriched the explanations in Sections 4 and 5.3, and included detailed pseudocode to better illustrate the model and its scalability. We also provided runtime plots and additional discussions to address concerns about computational efficiency and memory usage. Furthermore, we explored potential applications and datasets where DMD-GNN could demonstrate unique advantages, as well as scenarios where it could excel beyond traditional GNN architectures.
>
> We deeply appreciate your supportive review and the time you invested in providing such insightful feedback. Your encouragement motivates us to continue refining our work and exploring the potential of DMD-GNN further. Thank you once again!

---

> > ### Author Response · Authors · 2024-11-22
> > **Detailed Response Part 1**
> >
> > **Comment: It would be wise of the authors to attempt to preempt this confusion with generous use of intuitive figures. Perhaps an additional figure outlining arguments in section 4. Figure 1 covers some, but not all, of this.**
> >
> > **Reply:** Thank you very much for your kind suggestion. We agree that figures are important in helping to explain GNN dynamics and Koopman theory or DMD. Therefore, in our revised submission, we have enriched the last paragraph of Section 4 to make the relevant contents more explicit. Moreover, we refined the last paragraph in Section 5.3, **Summary of the Model Procedure** to provide a more detailed illustration of how our model is constructed. Lastly, in Appendix C.1, we show the **pseudocode** of both the DMD algorithm and our DMD-GNNs.
> >
> > **Comment: Additionally, the authors claim scalability by showing an experiment on 'OGB-arXiv', but I would like a cleaner argument for scalability. It seems a power (s=2) of the adjacency is used in some experiments. This can present issues for memory and runtime.**
> >
> > **Reply:** Thank you very much for pointing this out. The scalability of our method is ultimately determined by the DMD update in (14). To make this clearer, we added the pseudocode in Algorithms 1 and 2 in Appendix C. DMD-GNN has three major components in terms of computation: GNN, DMD, and spectral filtering. Apparently, DMD-GNN scales with the chosen GNN. DMD is a one-time computation and essentially an SVD where efficient methods such as [1] can be applied. The spectral filtering in (14) has similar scalability as a spectral GNN, although $\boldsymbol{\theta}\in\mathbb R^r$ and $r<\min(N,d)$ thanks to the low rank in DMD. To summarize, the scalability of DMD-GNN is mainly determined by the GNN used and a spectral GNN. To make it clear, we have made corresponding contextual changes on P8 and P19 in the revision.
> >
> > In regards to $s$, we fix $s =2$ followed by the settings in the SGC original paper [2]. The computation of $\widehat{\mathbf A}^s$ is one time and hence can be precomputed to save time.
> >
> > Finally we point out that one may need to apply training strategies such as [3] and [4] that are designed specifically for fitting large-scale graph datasets. Adopting this may lead to a potential for applying DMD-GNN to analyze those dynamics with large-scale graph. However, this is out of the scope of this paper since our focus is to demonstrate the effectiveness of DMD to enhance a GNN.
> >
> > [1]: Double Nyström Method: An Efficient and Accurate Nyström Scheme for Large-Scale Data Sets
> >
> > [2]: Simplifying Graph Convolutional Networks.
> >
> > [3]: Cluster-gcn: An efficient algorithm for training deep and large graph convolutional networks.
> >
> > [4]: Training Graph Neural Networks with 1000 Layers
> >
> >
> > **Comment:  I would also like to see some actual runtime plots/figures/numbers. Don't feel the need to re-run everything to record the times. Just enough to give the reader a sense of scale; what will it cost me (time, memory, etc) to actually run this?**
> >
> > **Reply:** Thank you very much for your suggestion. In Appendix C.2, we include the experiment runtime (DMD-GCN vs GCN(two layers)) of the node classification in citation networks (Figure 3); in addition, we also include the information on the actual ranks used in DMD-GNNs via different datasets (Table 5). Due to the time limit, it was quite challenging for us to include more figures and tables regarding the experiment statistics. We are happy to include them at any further stage.

---

> ### Author Response · Authors · 2024-11-22
> **Detailed Response Part 2**
>
> **Question: I am really looking for a setting that can show the unique benefit of DMD-GNN. The authors have gone through extensive derivations and work to derive DMD-GNN. What can it do that a generic, off-the-shelf, large GNN simply cannot do? What tasks does it make significantly easier (perhaps beyond just the final metric going up a bit)? Can the authors think of any applications, and ideally datasets, for this purpose? Perhaps data generated from systems closer to how DMD/Koopman theory is typically used would be a nice direction.**
>
>
> **Reply:**  We thank you very much for this insightful question. As a fundamental tool for analyzing dynamic systems, we suppose that DMD shall also remain powerful via the dynamics in which GNN is involved. In particular, in areas such as dynamic prediction, the main task is to simulate and predict the particles' position and feature changes via a physical system [1], [2]. Many recently developed methods tend to construct the graph (e.g., for the molecules) via a combination of KNN and its original topology, and such graph connectivity remains unchanged through the rest of the simulation/prediction. Although many physical/chemistry inductive biases have been introduced, a data-driven approach that generates a suitable graph structure that fits the propagation defined in the relevant models remains insufficiently explored to the best of our knowledge. Accordingly, DMD may own its advantage in resolving this challenge. Furthermore, a deeper reason for this could refer to the changes in the compound structure due to the changes in its energy (e.g., Gibbs free energy). This alignment may lead to a deeper insight into the relationship between numerical data-driven methods in machine learning and actual chemical/physical laws. We also expect to conduct our research in this field in our future research.
>
> The potential datasets for the aforementioned tasks could be (but not limited to):
>
> 1. CMU Motion Capture dataset [3] for 3D human motion capturing.
>
> 2. MD17 [4] for molecular dynamic prediction.
>
> 3. Adk equilibrium trajectory dataset [5] for protein dynamic prediction.
>
> [1]: Equivariant Graph Neural Operator for Modeling 3D Dynamics.
>
>
> [2]: E(n) Equivariant Graph Neural Networks.
>
>
> [3]: CMU. Carnegie-mellon motion capture database.
>
> [4]: Machine learning of accurate energy-conserving molecular force fields.
>
>
> [5]: Molecular dynamics trajectory for benchmarking mdanalysis.
>
> Furthermore, based on our examples in Appendix C.4.2 and C.6.2, we suppose that our DMD-GNNs may have the potential to enhance the prediction models in infectious disease epidemiology and physics. Lastly, based on your suggestion, we have included the relevant contents in our newly included section, Appendix D; please kindly visit our revised submission. Thank you again.

---

### Official Review · Reviewer_3qZf · 2024-11-04

**Soundness:** 2
**Presentation:** 3
**Contribution:** 2
**Rating:** 6
**Confidence:** 3

**Summary:**

The paper explores the relationship between Graph Neural Networks (GNNs) and Dynamic Mode Decomposition (DMD), proposing DMD-GNN models that enhance GNNs' ability to capture complex interactions in graph data through low-rank approximations of dynamic operators. The authors establish a theoretical connection between GNNs and the Koopman operator via DMD, reducing the number of learnable parameters and computational complexity. Experimental results demonstrate that DMD-GNNs achieve superior performance on various tasks, including node classification and link prediction. This work highlights the potential of integrating DMD techniques into GNN frameworks to improve their performance and applicability.

**Strengths:**

1. The paper is well written and easy to follow, providing a thorough introduction to the backgrounds of GNNs and DMD. This clarity enhances the reader's understanding of the foundational concepts necessary for grasping the main contributions of the work.
2. Connecting DMD with GNNs is an interesting perspective.
3. The authors support their claims with extensive experiments demonstrating the effectiveness of the DMD-GNN models, including evaluations on directed graphs, large-scale graphs, long-range interactions, and spatial-temporal graphs.

**Weaknesses:**

1. My main question regarding the paper concerns the motivation: to what extent do the dynamics of GNNs actually influence their performance? Approaching this from the perspective of DMD is interesting, but what specific insights does it offer in understanding GNN performance?
2. On long-range graph datasets, DMD-GNNs appear to outperform traditional GNNs, but some of the baselines used in the paper seem relatively weak. As far as I know, there is a class of GNNs derived from optimization or energy function approaches that perform reasonably well on such datasets. It would be helpful if the authors could include a few examples of these models for comparison.
3. It could benefit from a more thorough comparison with frequency-based analysis methods. Since the modes identified by DMD correspond to specific patterns in dynamical systems—similar to how frequency-based methods assume learned graph signals contain both high-frequency and low-frequency components—exploring the relationship between these approaches could yield unique insights. Although the authors provide some explanations in Section 6, illustrating this comparison with specific datasets would strengthen their argument.

**Questions:**

I’m confused about the implementation details of the model, as the paper provides limited description in this part. Standard DMD requires explicit eigendecomposition, and I would like to know how $\Psi$ and $\theta$ are determined in Equation 14.

---

> ### Author Response · Authors · 2024-11-22
> **Summary of the Response to Reviewer 3qZf**
>
> We sincerely appreciate your thoughtful feedback and the time you dedicated to reviewing our work. Your comments have provided important directions for us to refine the manuscript. In our revisions, we have clarified the motivation of our study, improved the explanation of the connection between DMD and GNN dynamics, and introduced additional discussions and experiments to enhance the comprehensiveness of our comparisons. Furthermore, we provided a detailed pseudocode to address your concerns regarding implementation.
>
> We believe that we have strived to address your comments to the best of our ability. We hope these updates meet your expectations and provide clarity on the contributions of our paper. Thank you again for your valuable insights and constructive suggestions.

---

> > ### Author Response · Authors · 2024-11-22
> > **Detailed Response Part 1**
> >
> > **Comment: My main question regarding the paper concerns motivation: to what extent do the dynamics of GNNs actually influence their performance? Approaching this from the perspective of DMD is interesting, but what specific insights does it offer in understanding GNN performance?**
> >
> > **Reply:** We thank the reviewer for your valuable suggestion.  In terms of the insights that DMD can offer to analyze the GNN dynamics, although we have highlighted the role of DMD in analyzing GNN dynamics in our original Introduction section, we agree that an enriched, more explicit statement shall be included. Therefore, in our revised submission, we maximized our effort to revise the **second paragraph in the Introduction section** to ensure a better illustration of our motivation and functionality of the Koopman operator, DMD, and their impacts on GNN dynamics. Please refer to our revised submission for the details.
> >
> > In addition to the above revision, we included one extra subsection in Appendix (B.1) to provide additional discussion on the main motivation for developing different GNN dynamics and their functionalities. We also discuss the role of DMD in analyzing these dynamics in a simple, linear way, and our proposed DMD-GNNs can generate similar or better learning accuracy than their original counterparts. We hope our effort can meet your expectations and strengthen this weakness. Please let us know what else we can do to make our paper better. Thanks again.
> >
> > **Comment: On long-range graph datasets, DMD-GNNs appear to outperform traditional GNNs, but some of the baselines used in the paper seem relatively weak. As far as I know, there is a class of GNNs derived from optimization or energy function approaches that perform reasonably well on such datasets. It would be helpful if the authors could include a few examples of these models for comparison.**
> >
> > **Reply:** Thank you very much for your insightful observation. We highlight that our proposed DMD-based GNNs are not specifically designed to accomplish long-range graph datasets but rather are a generic way of enhancing GNNs in various tasks. Nevertheless, we still observe that DMD-GNNs show comparable or even better results than our included baselines. To resolve your concern, we have included one more discussion section in our revised submission in Appendix C.3.5. by stating the difference between our proposed model and models that specifically target on long-range graph datasets such as [1,2].
> >
> > [1]: FOSR: First-order Spectral Rewiring for Addressing Oversquashing in GNNs.
> >
> > [2]: Understanding Oversquashing and Bottlenecks on Graphs via Curvature.
> >
> > **Comment: It could benefit from a more thorough comparison with frequency-based analysis methods. Since the modes identified by DMD correspond to specific patterns in dynamical systems—similar to how frequency-based methods assume learned graph signals contain both high-frequency and low-frequency components—exploring the relationship between these approaches could yield unique insights. Although the authors provide some explanations in Section 6, illustrating this comparison with specific datasets would strengthen their argument.**
> >
> > **Reply:** Thanks for this great suggestion. In our original submission, we only compared our model with graph framelets [3], which is a multi-scale spectral GNN that learns the filtering functions on both low and high frequency domains. Based on your comments, in our revision, we further compared our model with one of the fundamental spectral GNNs, namely ChebNet [4]. In addition, we also include the discussion between the results of ChebNet and DMD-GNNs in the experiment result paragraph via our revised submission as follows:
> >
> > *Finally, one can observe that DMD-GNNs also outperform spectral GNNs such as ChebNet, suggesting that the produced from data-driven DMD (i.e., $\boldsymbol{\Psi}$ in equation (14) ) effectively enhances original GNN dynamics by offering a refined spectral domain.*
> >
> > We hope our revision meets your expectations; thanks again.
> >
> > [3]: How framelets enhance graph neural networks
> >
> > [4]: Convolutional Neural Networks on Graphs with Fast Localized Spectral Filtering

---

> ### Author Response · Authors · 2024-11-22
> **Detailed Response Part 2**
>
> **Question: I’m confused about the implementation details of the model, as the paper provides a limited description in this part. Standard DMD requires explicit eigendecomposition, and I would like to know how $\boldsymbol{\Psi}$
> and $\boldsymbol{\theta}$ are determined in Equation 14.**
>
> **Reply:** Thank you very much for pointing this out. From equation (14), the construction of $\boldsymbol{\Psi}$ indeed requires SVD of the feature matrix $\mathbf H(\ell)$ and the eigenvectors of $\mathbf K$. Following your suggestion, we include a new section in Appendix C.1, namely the **Pseudocode** of DMD-GNN in which we explicitly show the process of implementing DMD-GNN. We have also mentioned this in the main body of our paper. Due to the page restriction, we kindly ask you to go to Appendix C.1 in our revised submission. Thanks again.

---

> > ### Author Response · Authors · 2024-11-22
> > **Consider Increasing the Score**
> >
> > We sincerely hope that the revisions we have made address your comments and enhance the clarity and depth of the paper. In light of these efforts and the updates made, we kindly ask you to consider increasing our score. Your constructive feedback has been invaluable in strengthening our work, and we remain committed to improving the paper based on your guidance. Thank you again for your thoughtful review.

---

> > > ### Comment · Reviewer_3qZf · 2024-11-26
> > >
> > > Thank you for your response. Your reply has addressed most of my concerns. My major concern, however, remains regarding the motivation. From my perspective, in 2024, GNN dynamics does not seem to be a critical issue (though this might be my personal bias). While incorporating DMD into the analysis of GNN dynamics is indeed interesting, it is difficult to conclude that the performance improvement is due to a deeper understanding of GNN dynamics. The explanation in Appendix B.1 partially alleviates my concern.
> > >
> > > Overall, I find the paper structurally sound and complete in its analysis and presentation. Connecting DMD with GNNs is an interesting perspective. I will raise my score from 5 to 6. Best of luck!

---

> > > > ### Author Response · Authors · 2024-11-26
> > > > **Thank you very much for your effort!**
> > > >
> > > > Dear Reviewer:
> > > >
> > > > Thank you very much for your reply. We appreciate your efforts and time in helping us improve our paper!
> > > >
> > > > Sincerely, The Authors

---

### Official Review · Reviewer_RmcR · 2024-11-05

**Soundness:** 3
**Presentation:** 2
**Contribution:** 2
**Rating:** 5
**Confidence:** 3

**Summary:**

It presents an approach that integrates Dynamic Mode Decomposition (DMD) with Graph Neural Networks (GNNs), resulting in DMD-GNN models. These models are designed to capture the principal components driving the underlying complex physics on graphs, thereby enhancing feature propagation and reducing computational costs. The paper also explores the potential of incorporating additional constraints on DMD and deploying physics-informed DMD for directed graphs, indicating a range of future research directions at the intersection of DMD and GNNs.

**Strengths:**

By integrating DMD with GNNs, the paper provides a new perspective for capturing dynamic patterns in graph data, which could be important for understanding dynamic behaviors in complex network structures.

**Weaknesses:**

The main proof is very similar to the proof in "Data-Driven Linearization of Dynamical Systems."

Although the paper proposes the DMD-GNN model, it does not explicitly elaborate on the relationship between the Koopman operator and graph neural networks in the first chapters, which may affect the reader's understanding of the theoretical foundation and the perceived innovativeness of the model.

Although the paper points out the DMD-GNN's application in multiple learning tasks, it does not discuss the model's potential and challenges in specific fields (such as bioinformatics, social network analysis, etc.).

The experimental results are not convincing; for example, in Table 1, some of the best results are close to simple MLP's performance. Besides, it only solves some very traditional classification problems in the experiments.

The analysis of the new approach on graph subclasses (e.g., sparse, dense, small-world, etc.) is insufficient.

Summary of the Model Procedure: From Line 291 to Line 293 is unclear.

**Questions:**

Eq(12) and Line 237, which is correct? K = ?

What is the W(l) in Eq(14)?

Line 280: Could you please elaborate on the usage of the rate?

You explained what o() is in Eq (15), but what is Df(0)?

In Lemma 1, what is X? should be X(l)?

---

> ### Author Response · Authors · 2024-11-15
> **Further Clarification of the Comments**
>
> Dear Reviewer:
>
> Thank you so much for your comments on our paper. We are working hard to address your questions and comments. To properly answer all your comments/questions, we here kindly ask you to provide further clarification on the comments below:
>
>
> **The analysis of the new approach on graph subclasses (e.g., sparse, dense, small-world, etc.) is insufficient.**
>
> Thanks again for helping us make our paper better.
>
> Authors

---

> > ### Comment · Reviewer_RmcR · 2024-11-21
> > **graph subclasses**
> >
> > I would like to know (theoretically, and not necessarily in a very rigorous sense) under what graph properties the new method performs well. One motivation for this question is that, even from experiments alone, it is evident that the new method is not always the best; when it is not the best, the gap to optimality can be significant. Moreover, even when it is the best, the difference compared to suboptimal methods is not substantial. Do certain properties of the graphs determine these outcomes? It would be great if we could determine whether to use the new method based on whether the graph satisfies certain properties (naturally forming a subclass).

---

> ### Author Response · Authors · 2024-11-22
> **Summary of the Response to Reviewer RmcR**
>
> We sincerely thank you for your thoughtful and detailed feedback, which has been invaluable in refining our paper. We carefully considered and addressed each of your comments. Specifically, we clarified the relationship between the Koopman operator and GNN dynamics, improved theoretical explanations, and expanded discussions on the potential applications and challenges of DMD-GNNs. We also revised unclear notations, added pseudocode, and ensured consistency throughout the manuscript.
>
> We greatly appreciate your follow-up clarification on Comment 1.5, which has further guided our revisions. While some aspects require future exploration, we believe our updates reflect meaningful improvements. Thank you again for your constructive insights, and we hope our efforts address your concerns and meet your expectations.

---

> ### Author Response · Authors · 2024-11-22
> **Detailed Response Part 1**
>
> **Comment: The main proof is very similar to the proof in "Data-Driven Linearization of Dynamical Systems."**
>
> **Reply:**  We thank the reviewer for your comments. In our paper, we mentioned that our analysis of how DMD captures the underlying geometry of the dynamic is adapted from work [1] with similar settings.  In addition,  Lemma 1, included in the main part of the paper, has an illustrative purpose, and our proof serves as a simple extension of the main Theorem in [1].
>
> [1]: Data-Driven Linearization of Dynamical Systems.
>
> **Comment: Although the paper proposes the DMD-GNN model, it does not explicitly elaborate on the relationship between the Koopman operator and graph neural networks in the first chapters, which may affect the reader's understanding of the theoretical foundation and the perceived innovativeness of the model.**
>
> **Reply:** We appreciate your valuable comments and note that solid mathematical illustrations are needed to elaborate explicitly on the relationship between the Koopman operator and the GNN dynamics, for example, in sections 3 and 4 in our original submission. In addition, in the last paragraph, lines 213-215 in section 4, we have shown the analogies between discrete/continuous GNN dynamic systems and the system that we used to induce the Koopman operator.
> However, based on your comments, we have made the following additional refinements to illustrate the relationship between the Koopman operator and GNN below.
>
>
> 1. We have refined the **last paragraph of section 4** as follows:
>
> *Clearly, equation (6) can be seen as a special case of discrete dynamical system equation (3). Similarly, the diffusion processes, such as equation (7) and equation (8), are described as the continuous dynamical system equation (1). These analogies show the potential of linking the Koopman operator induced under the settings of equation (3) or equation (1) to the GNN dynamics.  In the next section, we will show how the Koopman operator theory and its numerical approximation method (i.e., DMD) can facilitate analyzing GNN dynamics.*
>
>
> 2.  We have included some additional clarifications in the **Introduction section**. For example:
>
> * **Explicit motivation for using DMD on GNNs:**  *From the dynamical systems perspective, a carefully analyzed and refined dynamic can potentially enhance GNN performance by providing deeper insights into feature propagation over graphs. However, to tackle more challenging tasks, recent GNNs often adopt advanced physical dynamics [2], increasing complexity and hindering interpretability. Thus, a well-established tool is needed to analyze these dynamics and uncover their deeper insights.*
>
> * **Illustration on the functionality of the Koopman operator, DMD, and their impacts on GNNs:**  *Specifically, DMD estimates the Koopman operator by offering its finite-dimensional representation (e.g., matrix) only through leveraging the states (i.e., *snapshots*) of the system, regardless of its complexity. In addition, DMD produces the so-called *DMD-modes*, which are a collection of refined eigenbases of the system's original operator (e.g., graph adjacency matrix) with potentially lower dimensions. These modes capture the principal components driving the underlying GNN dynamics and offer a data-driven adjusted domain for spectral filtering. Building on these advantageous properties of DMD, in this work, we propose DMD-enhanced GNNs (DMD-GNNs). We show that our proposed DMD-GNNs not only can capture the characteristics of the original dynamics in a linear manner but also have the potential to adopt more complex tasks (e.g., long-range graphs [3]) in which the original model of DMD-GNNs usually delivers weak performance.*
>
> [2]: From continuous dynamics to graph neural networks: Neural diffusion and beyond.
>
> [3]: Long range graph benchmark.
>
> We sincerely hope our effort in clarifying our paper can meet your expectations on this matter. Thanks again.
>
>
> **Comment: Although the paper points out the DMD-GNN's application in multiple learning tasks, it does not discuss the model's potential and challenges in specific fields (such as bioinformatics, social network analysis, etc.).**
>
>
>  **Reply:** Thank you very much for your insightful suggestion. Our experiments demonstrated some of the applications using publicly available data sets. To further strengthen it, in our revised submission, we have included a detailed discussion on the challenges and potential future directions of our model in various areas such as molecular dynamics, classic (quantum) many-body problems, etc. Please kindly visit Appendix D, particularly the paragraph **Adopting DMD to More Complex Dynamics** for more details. Also, please let us know if you need more information from us. Thanks again for making our paper better.

---

> ### Author Response · Authors · 2024-11-22
> **Detailed Response Part 2**
>
> **Comment: The experimental results are not convincing; for example, in Table 1, some of the best results are close to simple MLP's performance. Besides, it only solves some very traditional classification problems in the experiments.**
>
> **Reply:** Thank you very much for your suggestion. We highlight the reason that some outputs in Table 1 are closed to MLP since MLP often serves as one of the state-of-the-art models for heterophilic graphs in which the graph topology (i.e., connectivity) is less functional than homophilic graphs where connected nodes are often with the similar labels. In addition, in our original submission, we have provided many empirical studies other than node classification tasks in our paper. For example, we show the performances of our model for link prediction in which our model is used as a graph encoder in section 7.5; we show the power of our model in terms of the spatiotemporal graph (STG) for spatiotemporal dynamic prediction in section 7.3. Furthermore, in our original submission, we show the potential of incorporating our model to enhance the models for infectious disease epidemiology in Appendix C.3.2 (now Appendix C.4.2 in the revised version), and we mentioned in section 7.3 of the main page) and briefly discuss our model's power to solve quantum time-dependent Schrödinger equation in Appendix C.5.2 (Now Appendix C.6.2). All these examples suggest that our model can be adapted to many other tasks rather than classification.
>
> **Comment: The analysis of the new approach on graph subclasses (e.g., sparse, dense, small-world, etc.) is insufficient.**
>
> **Reply:** We thank the reviewer for the intriguing question and acknowledge that providing a theoretical answer to such a question is extremely challenging. Indeed, the relations between graph properties and the GNN are intricate in pursuing given task performance, for example, connectivity and edge weights, nodes feature distribution, GNN structure (number of layers, type of convolution type, etc.), and optimization details. To show an example, let us recall the over-smoothing problem, as one may have a similar question on whether a more sparse graph has less risk of having the over-smoothing problem in terms of a specific GNN.  For example, one can have an unweighted, very dense graph (e.g., complete), while node features are very distinguished from each other. Then, running a GNN model, e.g., GCN on this complete graph, may not lead to the over-smoothing problem via the first several layers. On the other hand, one can also have a very sparse graph in which most features are similar to each other. In this case, running a GCN will soon lead to an over-smoothing problem.
>
> Based on the above statement, the learning accuracy of GNN models depends on not only graph connectivity but also node features. This is one of the reasons why the GNN research considers information on both connectivity and feature variation, such as the so-called Dirichlet energy with the form of $\mathrm{Tr}(\mathbf X^\top \mathbf{L} \mathbf X) = \sum_{i,j \in \mathcal E} w_{i,j}(\mathbf x_j - \mathbf x_i)^2$. Finally, finding a set of specific graph properties that are best suited for our dynamic (i.e., DMD-GNNs) is an interesting but highly non-trivial question, especially when our method is data-driven (i.e., depends on the node feature propagation). Similar difficulty can be found when research is conducted via other data-driven attention models such as GAT. As such, we reckon that the answer to your question might require control in a graph structure, including the distribution of the node features and the criteria for generating edges. One possible way of doing this is through the so-called Contextual Stochastic Block Model (CSBM). Then, further analysis can be conducted on this controlled graph. Again, we thank the reviewer for this interesting question, and we will keep seeking the answers to this question.

---

> > ### Author Response · Authors · 2024-11-22
> > **Detailed Response Part 3**
> >
> > **Comment: Summary of the Model Procedure: From Line 291 to Line 293 is unclear.**
> >
> > **Reply:** Thank you very much for pointing this out. To address your concern. We have refined the paragraph **Summary of the Model Procedure** below:
> >
> > *We visualize the steps on how our DMD-GNNs are designed in Figure 1. Initially, the underlying graph features $\mathbf x$ inside the top left blue circles are measured by the measurement function $\boldsymbol{\Phi}$, leading to the observed features $\mathbf h$ (in the left bottom orange cycles), which is further propagated by the initial GNNs to supply the inputs (snapshots) to DMD. DMD then takes these snapshots, e.g., $\mathbf h(\ell)$ and $\mathbf h(\ell +1)$, to produce the low-rank estimation of the GNN dynamics, namely DMD modes. After a customized truncation, the DMD modes (e.g., $\boldsymbol{\Psi} \in \mathbb R^{N\times r}$) are leveraged via DMD-GNNs to propagate node features via spectral filtering.*
> >
> > In addition, in our revised submission, we also include the pseudocode of our methods in Appendix C.1 as further clarification.
> >
> > **Question: Eq(12) and Line 237, which is correct? K = ?**
> >
> > **Reply:** We thank the reviewer for pointing this out. We apologize that we missed one $\mathbf M$ at line 237; the correct expression shall be the one in equation (12). In our revised submission, we have removed the expression of $\mathbf K$ at line 237 to prevent any further confusion.
> >
> > **Question: What is the $W(\ell)$ in Eq(14)?**
> >
> > **Reply:** We apologize for  missing the statement for $\mathbf W(\ell)$ in equation (14). We have revised our submission by denoting $\mathbf W(\ell)$ as a learnable matrix for channel-mixing.
> >
> > **Question: Line 280: Could you please elaborate on the usage of the rate?**
> >
> > **Reply:** Thanks for the suggestion. The usage of $\xi$ is to adjust the degree of truncation via the SVD in DMD. We suppose that your confusion is due to a lack of clarification. Accordingly, we have refined our statement in our revised submission.
> >
> > *Specifically, a rate $\xi \in [0,1]$ is assigned to control the rate of truncation via the SVD process in DMD, e.g., $\xi = 0.85$ means
> > DMD will select the singular values (from the largest) that are needed to reach 85\% of the total spectral energy.*
> >
> > Followed by some additional clarification. Please visit our revised submission for more details. In addition, in Appendix C, we have visualized (Figure 4) the utilization of $\xi$ in terms of the truncation rate adjustment and the actual ranks (Table 5) that are used in DMD-GNNs via different graphs. We hope our efforts can resolve your concern. Thanks again.
> >
> > **You explained what o() is in Eq (15), but what is Df(0)?**
> >
> > **Reply:** Thank you very much for pointing this out. In our revised submission, we have added the following clarification:
> >
> > *We also let $D\mathbf f(\boldsymbol{0})$ be the Jacobian matrix of the function $\mathbf f(\mathbf x)$ when $\mathbf x = \boldsymbol{0}$.*
> >
> >
> > **Question: In Lemma 1, what is X? should be X(l)?**
> >
> > **Reply:** We sincerely thank the reviewer for pointing this out. You are definitely correct, and we dropped $\ell$ in our original submission to avoid cluttered notations. In our revised paper, we have adequately refined the notation in both the informal and formal versions of Lemma 1 on the main page and Appendix.

---

> ### Author Response · Authors · 2024-11-22
> **Score Recosideration**
>
> We hope that we have addressed all of your comments thoroughly within the limited time available and made meaningful improvements to the manuscript. If there are any remaining concerns, we would greatly appreciate further clarification so that we can continue to refine the paper.  Meanwhile, we kindly ask you to reconsider increasing your score based on our detailed responses and the updates we made. Thank you again for your thoughtful and constructive feedback, which has been invaluable in helping us improve our work.

---

> > ### Author Response · Authors · 2024-11-25
> > **Follow-up Response**
> >
> > Dear Reviewer,
> >
> > Thank you for taking the time to provide your thoughtful and constructive feedback on our paper. We deeply appreciate your efforts and have carefully considered your comments and suggestions.
> >
> > While some of your suggestions, such as identifying a specific graph type most suitable for our method or extending the methodology to biological domains, are beyond the immediate scope of this work, we have included a discussion on these directions as potential future studies over here and our revised submission (e.g., Appendix D). We hope this provides clarity on how these ideas might be explored further in future research.
> >
> > We have also worked to address your other concerns to the best of our ability, as we believe they are important for strengthening the clarity and contribution of our work, especially in establishing a connection between modern Koopman learning theory (via dynamic mode decomposition) and graph neural networks, which we hope you find compelling.
> >
> > If you have any additional comments or further suggestions, we would be grateful to hear them and will be happy to address them promptly. Thank you again for your time and thoughtful engagement with our work.
> >
> > Sincerely,
> > The Authors

---

> > ### Comment · Reviewer_RmcR · 2024-11-26
> >
> > Thank you for your response; it has clarified many of my concerns. I understand that some aspects will need to be addressed in the future. However, the two main points I’m still worried about seem unavoidable: 1. The theoretical contribution is too similar to existing work technologically; 2. The experimental results are not particularly impressive.
> >
> > I’m willing to increase your scores for your efforts, but the points mentioned indicate a limit to how high I can go. I look forward to your future work and wish you all the best!

---

> ### Author Response · Authors · 2024-11-26
> **Thank you very much!**
>
> Dear Reviewer:
>
> Thank you very much for your time and patience in making our paper better. We will conduct future studies based on your comments and suggestions accordingly. Best of luck with your research as well!
>
> Sincerely, The authors.

---

### Author Response · Authors · 2024-11-22
**Response to Area Chair and All Reviewers**

We sincerely thank the Area Chair and Reviewers for their patience, insightful feedback, and constructive suggestions. We have carefully reviewed every comment and addressed them to the best of our knowledge and ability. Your valuable feedback has significantly helped us to refine and improve our work.

We have made revisions to the manuscript (highlighted in blue) in response to the points raised, including clarifying the motivation and theoretical foundations, refining our explanations of model design and implementation, addressing experimental concerns, and incorporating new discussions to enrich the context and highlight the broader applicability of our approach. Specific responses to individual comments, along with detailed explanations of the changes made, can be found in our responses to each reviewer.

We are deeply grateful for your thoughtful observations, which have allowed us to strengthen the paper and improve its presentation. Please feel free to let us know if there are further areas where we can refine our work. Thank you once again for your valuable input.

---

### Meta-Review · Area_Chair_7wVq · 2024-12-17

**Metareview:**

The main contribution of this paper is connecting GNNs with DMD.
Some reviewers claim that the paper's contributions are modest (the contribution is too similar to existing work), but the paper is well-written and it is interesting to connect these topics. There is no consensus on the appropriateness of the experiments. Some reviewers claim that the experiments are not particularly impressive, while others claim that the paper has extensive experiments demonstrating the effectiveness of the DMD-GNN models.

This paper is borderline.

**Additional Comments On Reviewer Discussion:**

It is worth noticing that two reviewers had low confidence in their review, so I weighted them lower in my decision.

---

### Decision · Program_Chairs · 2025-01-22

Accept (Poster)